# InfiMed: Low-Resource Medical MLLMs with Advancing Understanding and Reasoning

## Abstract

Multimodal Large Language Models (MLLMs) have achieved remarkable progress in domains such as visual understanding and mathematical reasoning. However, their application in the medical domain is constrained by two key challenges: (1) multimodal medical datasets are scarce and often contain sparse information, limiting reasoning depth; and (2) Reinforcement Learning with Verifiable Rewards (RLVR), though effective in general domains, cannot reliably improve model performance in the medical domain. To overcome these challenges, during the supervised fine-tuning (SFT) stage, we incorporate high-quality textual reasoning data and general multimodal data alongside multimodal medical data to efficiently enhance foundational medical capabilities and restore the base model's reasoning ability. Moreover, considering that there are some multimodal medical datasets with sparse information, we further synthesize reflective-pattern-injected chain-of-thought (CoT) in addition to general CoT samples, equipping the model with initial reflective reasoning capabilities that provide a structured foundation for subsequent RLVR training. Finally, we introduce our InfiMed-Series models, InfiMed-SFT-3B and InfiMed-RL-3B, both of which deliver state-of-the-art performance across seven multimodal medical benchmarks. Notably, InfiMed-RL-3B achieves an average accuracy of 59.2%, outperforming even larger models like InternVL3-8B, which achieves 57.3% Specifically, during the SFT phase, we utilized 188K samples, while the RLVR phase incorporated 36K samples, demonstrating the efficacy of both training strategies in achieving superior performance. We also conducted a series of extensive experiments, which provide valuable insights that contribute to advancing the performance of MLLMs in medical scenarios.

## 1 Introduction

The rapid development of multimodal large language models (MLLMs) in recent years has marked a transformative phase in artificial intelligence, driving substantial progress across diverse domains. Notably, MLLMs have achieved significant breakthroughs in areas such as object recognition (Yin et al., 2025; Liu et al., 2025d), mathematical reasoning (Zhuang et al., 2025; Peng et al., 2024; Liu et al., 2025b), and graphical user interface (GUI) interaction (Liu et al., 2025a; Luo et al., 2025; Qin et al., 2025), largely attributable to the availability of abundant high-quality multimodal datasets. In contrast, the medical domain remains particularly challenging due to the scarcity of high-quality multimodal data, which severely limits the performance of MLLMs in medical scenarios.

To enhance the medical reasoning capabilities of MLLMs, prior work has primarily relied on large-scale, domain-specific supervised fine-tuning (SFT). For instance, LLaVA-Med (Li et al., 2023) directly utilizes the PMC-15M (Zhang et al., 2023b) dataset for medical concept alignment and instruction following. However, its performance is constrained by the inherent noise of the dataset and the limited amount of reasoning information it provides. Recent studies, such as MedGemma (Sellergren et al., 2025), collect larger and higher-quality medical datasets that cover both textual and multimodal modalities, aiming to further enhance the general medical capabilities of MLLMs. While SFT can be effective, it is highly data-intensive and mainly focuses on memorizing training data (Chu et al., 2025). Building on the success of DeepSeek-R1 (Guo et al., 2025), Reinforcement Learning with Verifiable Rewards (RLVR) has shown significant improvements in exploration and generalization for multimodal tasks (Zhang et al., 2025; Liu et al., 2025a;c). RLVR, which often includes a "cold-start" phase in (MLLMs)(Huang et al., 2025; Peng et al., 2025; Liu et al., 2025a), is also

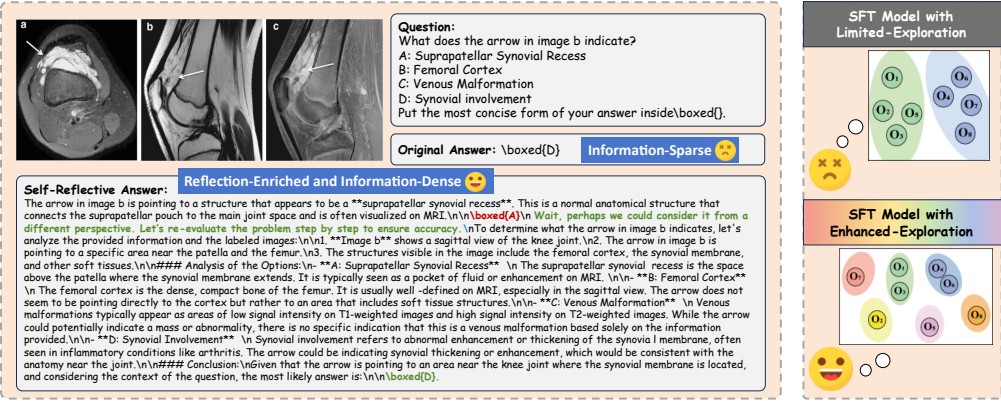

Figure 1: **Left:** Comparison of information-sparse and reflection-enriched, information-dense outputs. **Right:** A model with Enhanced Exploration (bottom) generates a broader, more effective search space, while Limited Exploration (top) results in a narrower, less efficient search.

beginning to find notable applications in medical scenarios (Su et al., 2025; Xu et al., 2025; Pan et al., 2025).

Despite these ongoing efforts, existing approaches still exhibit notable limitations, which can be summarized into two key challenges. First, the scarcity of high-quality multimodal medical datasets remains a bottleneck: most existing datasets suffer from sparse information and contain limited explanatory information, which hinders effective model training and results in poor reasoning performance, as shown in Figure 1. Second, although RLVR has been shown to substantially enhance model performance in other domains, its application in medical scenarios remains underexplored. Existing work either lacks extensive exploration across broad benchmarks (Pan et al., 2025; Su et al., 2025) or fails to effectively improve model performance (Xu et al., 2025).

To address the challenges mentioned above, during the SFT stage, we leverage not only multimodal medical data but also general multimodal data to preserve the model's visual perception capabilities, while integrating medical textual data to enhance its domain-specific knowledge. Additionally, we introduce a novel synthesis of reflective-pattern-injected chain-of-thought (CoT) data, effectively addressing the information sparsity present in certain multimodal medical datasets. This approach could also provide a more robust exploratory foundation for subsequent RLVR, enabling a cold-start method with limited resources. Building upon this, we train our InfiMed-SFT-3B model on 188K samples, equipping it with both fundamental reasoning and reflective patterns. We then apply RLVR on top of InfiMed-SFT-3B using 36K samples to obtain InfiMed-RL-3B, further enhancing its exploration capabilities and generalization performance. Extensive experiments show that our InfiMed-series models set new SOTA performance across multiple multimodal medical benchmarks, outperforming similarly-sized models like MedGemma-4B-IT and larger models such as InternVL3-8B, demonstrating the effectiveness of our reflective SFT and RLVR approach. We also investigated the impact of data composition and reasoning strategies through a series of exploratory experiments, yielding valuable insights for the advancement of medical MLLM applications.

In summary, the key contributions of our work are as follows: (**1**) We synthesize reflective-pattern-injected CoT data, equipping the model with initial reflective capabilities and a stronger cold-start foundation for subsequent RLVR. (**2**) We employ a low-resource SFT with 188K samples, enabling the model to develop robust reasoning, comprehension, and reflective patterns. Subsequently, RLVR is applied with 36K samples, effectively boosting the model's exploration capabilities and performance. (**3**) We introduce the InfiMed-series models, **InfiMed-SFT-3B** and **InfiMed-RL-3B**, which achieve SOTA performance among 3B-level MLLMs, with InfiMed-RL-3B outperforming models like MedGemma-4B-IT by 7.64%, and remain competitive even against 7B-level models.

## 2 RELATED WORK

### 2.1 MEDICAL MULTIMODAL LARGE LANGUAGE MODELS

In recent years, MLLMs have evolved rapidly and achieved remarkable progress across a wide range of domains, attracting increasing interest in their potential applications within the medical

field (AlSaad et al., 2024). Extensive research efforts have been devoted to enhancing MLLMs' ability to integrate heterogeneous medical data to support critical dimensions in healthcare. Inspired by the success of medical LLMs like HuatuoGPT (Zhang et al., 2023a), Apollo (Wang et al., 2024), and Med-PaLM series (Singhal et al., 2023; 2025), recent efforts have increasingly focused on extending LLM capabilities to multimodal medical. LLaVA-Med (Li et al., 2023) introduces a biomedical-specialized large language-and-vision model trained on a curated figure-caption dataset with self-instructed instruction-following data. The model highlights the potential of cost-efficient training strategies for domain-specific MLLMs. MedGemma (Sellergren et al., 2025) has shown strong generalization across medical vision-language and text-only tasks, demonstrating advanced medical understanding and reasoning on multimodal data. Lingshu (Xu et al., 2025) proposed a domain-specialized multimodal foundation model for medical, supported by a curated dataset enriched with medical VQA, CoT reasoning, and report annotations. While prior work has made notable progress in adapting MLLMs to the medical domain, many approaches depend on large model sizes and substantial computational resources, which limit their accessibility and scalability.

## 2.2 REASONING IN MEDICAL LARGE LANGUAGE MODELS

Interpretable reasoning remains a central desideratum in medical AI, with recent efforts exploring general CoT prompting (Wei et al., 2022) and program-based logic (Chen et al., 2022) modeling. Although these approaches have shown potential, they typically rely on costly expert-curated annotations (Li et al., 2024b), which limits their scalability in real-world clinical settings. RL offers a compelling alternative by enabling emergent reasoning capabilities without requiring explicit supervision, as demonstrated by recent models such as DeepSeek-R1 (Guo et al., 2025), which achieve notable improvements in reasoning with rule-based reward. Building on this paradigm, RLVR has been used to improve reasoning reliability, with Group Relative Policy Optimization (Shao et al., 2024) known for its efficiency and good performance. This method is now increasingly used to train MLLMs to improve their reasoning ability (Meng et al., 2025; Wang et al., 2025; Tan et al., 2025). With the success of RLVR, several work leverages it on medical MLLMs. MedVLM-R1 (Pan et al., 2025) employs RLVR to explicit reasoning in medical VQA, achieving strong performance and generalization. Its emphasis on reasoning highlights the role of RL in enhancing transparency and trustworthiness in clinical AI systems. GMAI-VL-R1 (Su et al., 2025) explores RLVR to enhance reasoning and reflection in multimodal medical models. By introducing a multi-agent reasoning data synthesis framework, the model outperforms prior models on some complex tasks. Lingshu (Xu et al., 2025) also leverages an RLVR paradigm, achieving strong performance across medical VQA, report generation, and text-only QA. Despite these promising advances, prior work has been limited in its exploration of the RLVR stage.

## 2.3 OUR DISTINCTION

As mentioned above, general CoT consists of multi-step natural language reasoning traces derived from instruction data. These traces teach the model structured reasoning patterns during SFT and help it select more appropriate responses during RLVR after learning task-solving behaviors. However, unlike open-domain tasks where many reasoning paths may be acceptable, medical reasoning is highly standardized and tightly constrained by clinical knowledge. This significantly limits the diversity of viable intermediate steps, reducing the exploration space available to RLVR and making it harder for the model to discover improved reasoning trajectories. To address this limitation, we introduce reflective-pattern–injected CoT data during SFT. This data provides the model with initial self-reflection and self-correction capabilities, effectively expanding the reasoning space that RLVR can explore and enabling more robust improvements on complex medical tasks. Moreover, existing studies either focus on a narrow set of benchmarks or fail to consistently improve the performance of the SFT model. In contrast, we not only successfully enhance model performance during the RLVR stage but also conduct extensive experiments to analyze the features in multimodal medical tasks.

## 3 METHODOLOGY

In this section, we outline our methodology for advancing multimodal medical understanding and reasoning through RLVR with a self-reflective cold start, which is depicted in Figure 2. Our approach unfolds in two stages: (1) A cold start phase, in which we uniquely integrate general

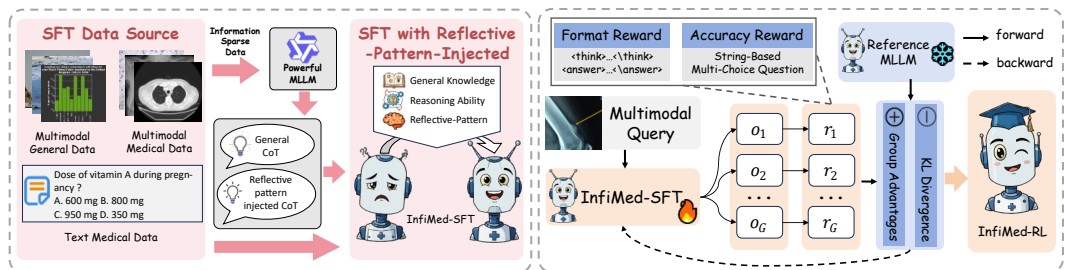

Figure 2: The overall training process of InfiMed-Series models.

multimodal data with medical text reasoning data to simultaneously enhance image understanding and restore fundamental reasoning skills. Crucially, to address the information sparsity of existing medical datasets, we further synthesize both distilled CoT and self-reflective CoT for SFT, thereby establishing a richer and more exploratory reasoning foundation. (2) A RLVR phase, which enables the model to explore a wider spectrum of reasoning trajectories, thereby producing more robust and clinically faithful multimodal reasoning.

## 3.1 REFLECTIVE-INJECTED SUPERVISED FINE-TUNING

As mentioned above, since SFT constitutes the foundation for subsequent RLVR, we incorporated not only general multimodal data but also text-based medical reasoning data during SFT to strengthen the model's fundamental multimodal understanding and reasoning capabilities (Sellergren et al., 2025; Xu et al., 2025). However, several existing multimodal medical SFT datasets suffer from insufficient informational richness. For instance, multiple-choice question datasets often only provide the final choice and always lack explicit explanation. To address this limitation, in addition to only generating conventional CoT data to supplement the missing information, we further construct reflective-pattern-injected CoT data, enabling the model to develop more comprehensive and self-corrective reasoning capabilities (Cheng et al., 2024).

The core premise of reflective-pattern-injected is that directly exposing the model to a spectrum of reasoning trajectories, including correct, partially inconsistent, and subtly flawed chains, encourages the development of self-evaluation and error-correction mechanisms.

Formally, given a multimodal medical question $q$, whose original response consists of insufficient information, consisting of a textual task instruction $x$ and one or more images $\mathcal{I}$, i.e. $q = \{x, \mathcal{I}\}$. We utilize a powerful MLLM (e.g., Qwen2.5-VL-32B (Bai et al., 2025)) to generate a batch of candidate responses $\{y_i\}$. Subsequently, leveraging rejection sampling, we partition these candidates into two disjoint subsets: $\{y_i^+\}$, corresponding to correct responses, and $\{y_i^-\}$, corresponding to incorrect responses. For the correct responses $\{y_i^+\}$, we further engage a more advanced MLLM (e.g., Qwen2.5-VL-72B (Bai et al., 2025)) to evaluate each response across multiple dimensions, including clinical accuracy, logical reasoning, factual correctness, and completeness.

Finally, we synthesize a reflective-pattern-injected CoT by combining one of the highest-quality responses from $\{y_i^+\}$ with a randomly selected response from $\{y_i^-\}$, thereby creating a novel training instance that emphasizes both reasoning and error-awareness. More details of the reflective-pattern-injected CoT synthesis can be found in the Appendix A.2.

## 3.2 REINFORCEMENT LEARNING WITH VERIFIABLE REWARDS

### 3.2.1 OVERALL PROCESS OF RLVR

After the SFT stage with reflective-pattern injection, we use Group Relative Policy Optimization (GRPO) in the RLVR phase to improve stability, building on the method from Deepseek-R1 (Guo et al., 2025). GRPO computes advantages by generating multiple responses for the same query, removing the need for an explicit critic model.

We formally denote the model after the SFT stage with reflective-pattern injection as $\pi_\theta$, the policy model in RLVR. Given a multimodal medical query $q$, the policy model $\pi_{\theta_{old}}$ (prior to parameter updates) generates a set of $G$ candidate responses $\{o_i\}_{i=1}^G$. For each response $o_i$, a rule-based

reward function $R(o, \text{gt})$ is used to evaluate its quality and assign a score $r_i$, where gt denotes the ground-truth answer. Based on the collection of rewards $\{r_i\}_{i=1}^G$, the group-relative advantages $\{A_i\}_{i=1}^G$, which quantify the relative quality of responses within the batch, can be calculated as:

$$A_i = \frac{r_i - mean(\{r_1, r_2, \ldots, r_G\})}{std(\{r_1, r_2, \ldots, r_G\})}, \tag{1}$$

where $mean(\cdot)$ indicates the average value, and $std(\cdot)$ refers to the standard deviation.

Based on the above group-relative advantages, GRPO updates the policy by maximizing the expected advantage-weighted likelihood ratio. The optimization objective can be formulated as:

$$\mathcal{J}_{\text{GRPO}}(\theta) = \mathbb{E}_{[q \sim P(Q), \{o_i\}_{i=1}^G \sim \pi_{\theta_{\text{old}}}(O|q)]} \frac{1}{G} \sum_{i=1}^G \frac{1}{|o_i|}$$

$$\sum_{t=1}^{|o_i|} \left\{ \min \left[ \frac{\pi_\theta(o_i|q)}{\pi_{\theta_{\text{old}}}(o_i|q)} A_i, \text{clip} \left( \frac{\pi_\theta(o_i|q)}{\pi_{\theta_{\text{old}}}(o_i|q)}, 1 - \epsilon, 1 + \epsilon \right) A_i \right] - \beta D_{\text{KL}} \left[ \pi_\theta \| \pi_{\text{ref}} \right] \right\}, \tag{2}$$

where the additional Kullback–Leibler term $D_{\text{KL}} \left[ \pi_\theta \| \pi_{\text{ref}} \right]$ is applied to penalize divergence from the reference policy model $\pi_{\text{ref}}$, thereby helping to maintain training stability.

### 3.2.2 RULE-BASED REWARD CONSTRUCTION

Considering the reward function $R(o, \text{gt})$ aims to guide the policy model to learn a suitable and correct reasoning trajectory, we design our total reward $R_{total}$, which integrates assessments of both output format correctness and accuracy:

$$R_{\text{total}}(o, \text{gt}) = w_{\text{format}} \cdot R_{\text{format}}(o) + w_{\text{acc}} \cdot R_{\text{accuracy}}(o, \text{gt}), \tag{3}$$

where $R_{\text{format}}(o)$ denotes the reward for the correctness of the output format and $R_{\text{accuracy}}(o, \text{gt})$ denotes the reward for the accuracy of the output $o$ relative to the ground-truth result. The non-negative coefficients $w_{\text{format}}$ and $w_{\text{acc}}$ serve as hyperparameters weighting the relative contribution of the two components, with $w_{\text{format}} + w_{\text{acc}} = 1$.

The format reward $R_{\text{format}}(o)$ assesses whether the output of the policy model $\pi_\theta$ satisfies the predefined format. Notably, $R_{\text{format}}(o) \in \{0, 1\}$, where $R_{\text{format}}(o) = 1$ if all specified format requirements are satisfied; otherwise, $R_{\text{format}}(o) = 0$. Specifically, it verifies two primary aspects:

- **Thinking Progress:** We evaluate whether the model correctly presents its reasoning process according to a predefined format. Specifically, the model may be required to encapsulate its reasoning process and final answer within designated tags.
- **Final Answer Format:** We examine whether the model outputs an explicit final answer, with particular attention to cases where the instructions related to query $q$ require such a response.

The accuracy reward $R_{\text{accuracy}}(o, \text{gt})$ evaluates the correctness of the model output $o$ relative to the ground truth of query $q$. Importantly, $R_{\text{accuracy}}(o, \text{gt})$ is defined only when the output meets the format constraint, i.e., $R_{\text{format}}(o) = 1$; otherwise, it is zero. This design ensures that the model generates well-structured outputs before being evaluated for correctness. When $R_{\text{format}}(o) = 1$, the computation of $R_{\text{accuracy}}(o, \text{gt})$ depends on the task-specific ground-truth format. The two main tasks' reward functions are as follows; others are presented in the Appendix A.3.

- **String-based Tasks:** For textual answers, $R_{\text{accuracy}}(o, \text{gt})$ is computed by normalizing both the model output and the ground truth (e.g., lowercasing, removing redundant spaces). This function evaluates the extracted answer from the output $o$, denoted as $o_{\text{ans}}$, by comparing it to the ground truth answer gt. We use the Jaccard function to measure the similarity between $o_{ans}$ and $gt$. The Jaccard function can be formulated as: $Jaccard(o_{\text{ans}}, \text{gt}) = \frac{|o_{\text{ans}} \cap \text{gt}|}{|o_{\text{ans}} \cup \text{gt}|}$.
- **Multiple-Choice Questions:** For tasks that require selecting an option from a predefined set, $R_{\text{accuracy}}(o, \text{gt})$ is calculated by directly comparing the model's extracted predicted answer, $o_{\text{ans}}$, with the correct ground truth option, gt. A match results in a reward of 1, while a mismatch yields a reward of 0.

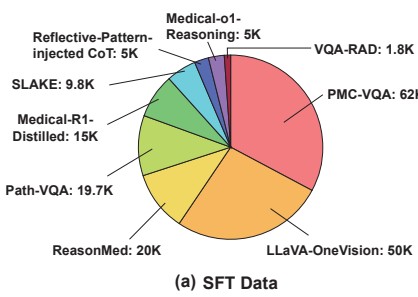 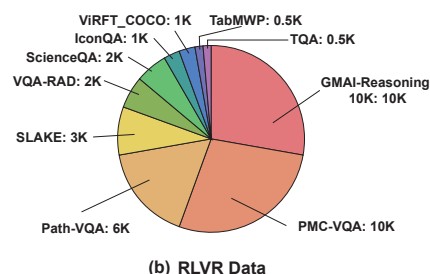

Figure 3: Overview of the training samples for the InfiMed series models in the reflective-injected SFT and RLVR stages.

## 4 EXPERIMENT

In this section, we present the experimental setup used to train and evaluate our proposed InfiMed-series models, which are built upon Qwen2.5-VL-3B-Instruct (Bai et al., 2025). We detail the implementation process, outline the evaluation benchmarks, and provide a comprehensive comparison with SOTA models. Furthermore, we analyse the impact of our training recipe to better understand its contributions to overall performance. We also aim to address the following research questions.

- **RQ1:** How do the InfiMed-Series models perform compared with other MLLMs across various medical benchmarks?
- **RQ2:** How do different data types and data numbers influence SFT and RLVR performance?
- **RQ3:** Does reasoning really enhance performance in medical tasks?
- **RQ4:** How do models trained with self-reflection via SFT compare to RLVR-optimized models in their quality and reliability of medical responses?
- **RQ5:** Can our SFT data consistently improve model performance across different base MLLM architectures?
- **RQ6:** Does increasing the amount of RLVR training data lead to further improvements in model performance?

### 4.1 EXPERIMENTAL SETUP

**Models.** We conduct a comprehensive comparison across a wide range of models. The models include: (1) Proprietary models: GPT-series models (Achiam et al., 2023), Claude Sonnet 4 (Anthropic, 2025), and Gemini-2.5-Flash (Comanici et al., 2025); (2) General open-source models: Qwen2.5-VL series models (Bai et al., 2025), Gemma3 series models (Team et al., 2025) and InternVL series models (Chen et al., 2024c; Zhu et al., 2025); (3) Medical open-source models: MedVLM-R1-2B (Pan et al., 2025), MedGemma-4B-IT (Sellergren et al., 2025), LLaVa-Med-7B (Li et al., 2023), HuatuoGPT-V-7B (Chen et al., 2024b), Lingshu-7B (Xu et al., 2025), BioMediX2-8B (Mullappilly et al., 2024).

**Datasets.** During the reflective-injected SFT stage, we utilize a total of **188K** samples from three categories: (1) multimodal general data, (2) multimodal medical data, and (3) text-based medical data. In the RLVR stage, we utilize **36K** multimodal medical datasets and multimodal general datasets. An overview of the training datasets is provided in Figure 3. The detailed description of the datasets is provided in the Appendix A.4.

**Evaluation Benchmark** We adopt seven widely used multimodal medical benchmarks: MMMU-Health&Medicine (MMMU-H&M) (Yue et al., 2024), VQA-RAD (Lau et al., 2018), SLAKE (Liu et al., 2021), PathVQA (He et al., 2020), PMC-VQA (Zhang et al., 2023c), OmniMedVQA (Hu et al., 2024), and MedXpertQA (Zuo et al., 2025). These benchmarks span a wide range of imaging modalities, including X-ray, CT, MRI, PET, ultrasound, and pathology. Collectively, they form a comprehensive evaluation framework for assessing both reasoning ability and proficiency in general medical knowledge. A detailed description of these benchmarks is provided in Appendix A.4.

Additional experimental setup details are provided in Appendix A.4.

Table 1: **Performance comparison of different MLLMs across various medical vision-language benchmarks.** Results of comparison of InfiMed-series models with other MLLMs on medical multi-modal benchmarks. MMMU-H&M, OMVQA, and MedXQA denote MMMU-Health&Medicine, OmniMedVQA, and MedXpertQA-Multimodal, respectively. The best results among models in the 2-4B parameter are **bolded**.

| Model | Size | Accuracy (%) | | | | | | | |
|---|---|---|---|---|---|---|---|---|---|
| | | MMMU-H&M | VQA-RAD | SLAKE | PathVQA | PMC-VQA | OMVQA | MedXQA | Avg. |
| *Proprietary Models* | | | | | | | | | |
| GPT-5 | - | 83.6 | 67.8 | 78.1 | 52.8 | 60.0 | 76.4 | 71.0 | 70.0 |
| GPT-5-mini | - | 80.5 | 66.3 | 76.1 | 52.4 | 57.6 | 70.9 | 60.1 | 66.3 |
| GPT-5-nano | - | 74.1 | 55.4 | 69.3 | 45.4 | 51.3 | 66.5 | 45.1 | 58.2 |
| GPT-4.1 | - | 75.2 | 65.0 | 72.2 | 55.5 | 55.2 | 75.5 | 45.2 | 63.4 |
| Claude Sonnet 4 | - | 74.6 | 67.6 | 70.6 | 54.2 | 54.4 | 65.5 | 43.3 | 61.5 |
| Gemini-2.5-Flash | - | 76.9 | 68.5 | 75.8 | 55.4 | 55.4 | 71.0 | 52.8 | 65.1 |
| *General Open-source Models* | | | | | | | | | |
| Qwen2.5VL-3B | 3B | 51.3 | 56.8 | 63.2 | 37.1 | 50.6 | 64.5 | 20.7 | 49.2 |
| Gemma3-4B | 4B | 34.0 | 49.9 | 61.1 | 43.2 | 47.9 | 60.9 | 20.9 | 45.4 |
| Qwen2.5VL-7B | 7B | 54.0 | 65.0 | 67.6 | 44.6 | 51.3 | 63.5 | 21.7 | 52.5 |
| InternVL2.5-8B | 8B | 53.5 | 59.4 | 69.0 | 42.1 | 51.3 | 81.3 | 21.7 | 54.0 |
| InternVL3-8B | 8B | 59.2 | 65.4 | 72.8 | 48.6 | 53.8 | 79.1 | 22.4 | 57.3 |
| *Medical Open-source Models* | | | | | | | | | |
| MedVLM-R1-2B | 2B | 35.2 | 48.6 | 56.0 | 32.5 | 47.6 | **77.7** | 20.4 | 45.4 |
| MedGemma-4B-IT | 4B | 43.7 | 49.9 | 76.4 | 48.8 | 49.9 | 69.8 | 22.3 | 51.5 |
| LLaVA-Med-7B | 7B | 29.3 | 53.7 | 48.0 | 38.8 | 30.5 | 44.3 | 20.3 | 37.8 |
| HuatuoGPT-V-7B | 7B | 47.3 | 67.0 | 67.8 | 48.0 | 53.3 | 74.2 | 21.6 | 54.2 |
| Lingshu-7B | 7B | 54.0 | 67.9 | 83.1 | 61.9 | 56.3 | 82.9 | 26.7 | 61.8 |
| BioMediX2-8B | 8B | 39.8 | 49.2 | 57.7 | 37.0 | 43.5 | 63.3 | 21.8 | 44.6 |
| *Ours (InfiMed-Series)* | | | | | | | | | |
| InfiMed-SFT-3B | 3B | 54.7 | 58.1 | 82.0 | 60.6 | 53.2 | 67.0 | 23.5 | 57.1 |
| Gemma3-SFT-4B | 4B | 35.3 | 59.9 | **83.3** | **64.7** | 53.3 | 68.7 | 21.0 | 55.2 |
| InfiMed-RL-3B | 3B | **55.3** | **60.5** | 82.4 | 62.0 | **58.7** | 71.7 | **23.6** | **59.2** |

## 4.2 RESULTS ON VARIOUS MEDICAL BENCHMARKS (RQ1 & RQ5)

Table 1 presents a comprehensive comparison of different MLLMs across seven diverse medical vision-language benchmarks. Among all models, proprietary closed-source models (e.g., GPT-5, Gemini-2.5, Claude) consistently outperform both general-purpose and medical-domain open-source models, achieving the highest average accuracy (e.g., 70.0% for GPT-5). These models set a strong upper bound, particularly excelling on complex benchmarks such as MMMU-H&M and MedXpertQA, indicating their superior reasoning and image understanding capabilities.

Furthermore, comparisons with existing open-source models show that the InfiMed-series models offer significant performance advantages. Both InfiMed-SFT-3B and InfiMed-RL-3B notably outperform other models of similar scale, achieving average accuracies of 57.1% and 59.2%, respectively, across seven multimodal medical benchmarks. We also notice that MedVLM-R1-2B achieves 77.7% on OmniMedVQA, primarily because its training dataset may overlap with a portion of the OmniMedVQA benchmark.

Notably, our 3B models outperform some larger 7B and 8B models, such as HuatuoGPT-V-7B and InternVL2.5-8B, despite their greater scale. Although a gap remains between InfiMed-RL-3B and Lingshu-7B, our model achieves competitive performance with fewer parameters and using a low-resource dataset (188K for SFT and 36K for RLVR), compared to Lingshu-7B's 12M samples, HuatuoGPT-V-7B's 1.3M samples, highlighting the efficiency and effectiveness of our training.

Although models such as MedVLM-R1-2B, GMAI-VL-R1-7B (not open-sourced), and Lingshu-7B provide some evidence that RLVR can be effective after SFT, they either target only a narrow range of benchmarks or fail to achieve consistent overall gains. By contrast, the 2.1% overall improvement of InfiMed-RL-3B over InfiMed-SFT-3B, along with consistent gains across seven medical benchmarks, clearly demonstrates that RLVR not only could enhance model performance in the medical domain but also complements our SFT phase training, thereby substantiating the effectiveness of RLVR.

To further evaluate the model-agnostic robustness of our data, we incorporated a fundamentally different model family, the Gemma3 series, and fine-tuned a Gemma3-4B-IT model using our SFT data. Despite the architectural and training differences between Gemma and previously evaluated models such as Qwen, our data still produces substantial performance gains over the original Gemma

Table 2: **Ablation study examining data composition during the training stage.** $\Delta|$Data$|$ denotes the amount of data change applied to the training set. w/o-general, w/o-text, and w/o-refcot refer to training configurations where the general multimodal data, textual medical data, and reflective-pattern-injected CoT data are removed, respectively. gen_mm, text, and general_cot denote the general multimodal data, medical textual data, and general CoT data components included in the training corpus.

| Model | $\Delta|$Data$|$ | Accuracy (%) | | | | | | | |
|---|---|---|---|---|---|---|---|---|---|
| | | MMMU-H&M | VQA-RAD | SLAKE | PathVQA | PMC-VQA | OMVQA | MedXQA | Avg. |
| *Base Model* | | | | | | | | | |
| Qwen2.5VL-3B | - | 51.3 | 56.8 | 63.2 | 37.1 | 50.6 | 64.5 | 20.7 | 49.2 |
| *Ablation Study in SFT Stage on General Multimodal Data* | | | | | | | | | |
| InfiMed-SFT-3B | - | 54.7 | 58.1 | 82.0 | 60.6 | 53.2 | 67.0 | 23.5 | 57.1 |
| InfiMed-SFT-3B+gen_mm | +20K | 54.0 | 60.7 | 81.8 | 55.8 | 55.5 | 67.5 | 22.3 | 56.8 |
| InfiMed-SFT-3B−gen_mm | −20K | 48 | 58.1 | 82.1 | 58.3 | 51.4 | 66.9 | 22.6 | 55.4 |
| *Ablation Study in SFT Stage on Medical Text Data* | | | | | | | | | |
| InfiMed-SFT-3B-w/o-general | −50K | 50.0 | 60.5 | 80.5 | 60.4 | 51.6 | 59.7 | 22.6 | 55.1 |
| InfiMed-SFT-3B+text | +20K | 50.7 | 63.4 | 80.4 | 57.3 | 54.4 | 67.3 | 21.4 | 56.4 |
| InfiMed-SFT-3B−text | −20K | 50.7 | 60.3 | 82.2 | 58.1 | 53.8 | 67.3 | 22.6 | 56.4 |
| InfiMed-SFT-3B-w/o-text | −40K | 44.0 | 61.0 | 81.6 | 60.4 | 51.1 | 64.7 | 21.9 | 54.9 |
| *Ablation Study in SFT Stage on Reflective CoT Data* | | | | | | | | | |
| InfiMed-SFT-3B-w-generalcot | − | 50.7 | 60.5 | 81.9 | 57.7 | 52.7 | 66.4 | 23.4 | 56.2 |
| InfiMed-SFT-3B-w/o-refcot | −5K | 50.0 | 60.1 | 81.3 | 60.5 | 53.1 | 64.7 | 22.8 | 56.1 |
| *Ablation Study in RLVR Stage* | | | | | | | | | |
| InfiMed-RL-3B | - | 55.3 | 60.5 | 82.4 | 62.0 | 58.7 | 71.7 | 23.6 | 59.2 |
| InfiMed-RL-3B-w/o-general | −10K | 53.3 | 60.7 | 81.9 | 61.6 | 58.3 | 70.0 | 23.6 | 58.4 |

baseline and achieves state-of-the-art performance within the 3–4B parameter range (excluding comparisons with InfiMed itself). Although this experiment relies solely on SFT, the strong improvements obtained with a relatively small dataset of 176K samples demonstrate that the effectiveness of our SFT data is not tied to any specific backbone family. These results provide compelling evidence that our data is highly efficient and generalizes well across heterogeneous MLLM architectures.

### 4.3 ABLATION STUDY ON DATA COMPOSITION (RQ2 & RQ6)

For the SFT and RLVR stage, we assessed the contribution of each component in our dataset by conducting an ablation study. For the SFT, we systematically remove specific types of training data, including general multimodal data, textual medical data, and reflective-pattern-injected CoT data. The overall detailed results are presented in Table 2. Based on these experiments, we draw the following conclusion: ***Unlike general multimodal tasks, medical multimodal problems are inherently comprehensive, requiring the integration of textual, visual, and domain-specific knowledge. As a result, medical training datasets alone are insufficient to ensure robust MLLM performance.*** A detailed analysis is provided below:

During the SFT stage, we observe that each data component serves a distinct function. Removing general multimodal data has a pronounced negative effect on benchmarks like OmniMedVQA, which require nuanced visual understanding. This suggests that general-domain multimodal examples help the model interpret complex visual patterns, align visual and textual features, and handle diverse image information in medical-specific datasets. Excluding textual medical data severely degrades performance on MMMU-H&M, indicating that such data provides critical domain-specific knowledge, including medical terminology, clinical reasoning strategies, and structured question-answering patterns essential for accurate interpretation and reasoning.

In addition to the ablations on each data component, we performed supplementary experiments to examine the effect of moderate changes in data proportions. We increased and decreased the amounts of general multimodal data and textual medical data by 20K samples. As shown in Table 2, these adjustments lead to performance differences across benchmarks, reflecting the complementary roles of the two data types. Our current data composition was determined based on empirical observations from preliminary experiments, which indicated that this setting offers a stable balance between visual understanding and medical-domain reasoning. Although the results suggest that alternative ratios may provide further gains, an exhaustive search for optimal proportions requires a broader investigation.

Moreover, even though reflective-pattern-injected CoT data contains only 5K examples, its removal leads to noticeable declines across most benchmarks, highlighting its role in enhancing multi-step reasoning and self-reflection for complex or ambiguous medical questions. Interestingly, VQA-RAD performance slightly increases after removing certain data, as this older benchmark emphasizes mem-

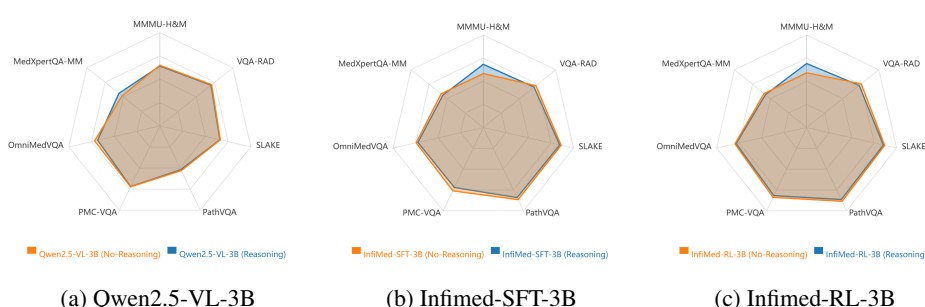

(a) Qwen2.5-VL-3B        (b) Infimed-SFT-3B        (c) Infimed-RL-3B

Figure 4: Comparison of direct-answer and reasoning-based prompts on medical benchmarks.

orization; reducing other data effectively increases the relative proportion of VQA-RAD examples, yielding a modest gain.

To further isolate the contribution of reflective CoT, we conducted an additional ablation in which we replaced the reflective CoT data with an equal amount of general CoT samples. The model equipped with reflective CoT consistently outperformed the one trained with general CoT, demonstrating that the reflective formulation itself provides unique benefits by guiding the model to articulate intermediate reasoning, identify potential errors, and refine its final predictions with greater reliability.

The lower half of Table 2 reports the RLVR ablation, focusing on variants excluding general multi-modal data. When removed, performance on MMMU-H&M drops below InfiMed-SFT-3B, suggesting that RLVR relying solely on medical multimodal data, which is typically less reasoning-intensive, reduces overall reasoning capability, leading to lower performance. More ablation studies are presented in the Appendix A.5.

### 4.4 ANALYSIS OF REASONING EFFECTIVENESS IN MEDICAL SCENARIOS (RQ3)

To assess reasoning effectiveness in medical scenarios, we evaluated the model using two prompts: (1) a direct-answer prompt, where the model is asked to output only the final prediction; (2) a reasoning-augmented prompt, where the model is encouraged to generate intermediate reasoning steps before providing the answer. This setup allows us to examine the impact of explicit reasoning. Results are shown in Figure 4.

Our experiments reveal a consistent trend: explicit reasoning prompts tend to reduce performance on most medical benchmarks, with two exceptions. First, both InfiMed-SFT and InfiMed-RL benefit from reasoning on MMMU-H&M. Second, the general-purpose Qwen2.5-VL-3B benefits on both MMMU-H&M and MedXpertQA-MM. This indicates that explicit reasoning alone does not universally improve medical-focused MLLMs, even with additional optimization like RLVR. We attribute the gains on MMMU-H&M and MedXpertQA-MM (for Qwen2.5-VL-3B) to their reasoning-intensive design requiring multi-step logical deduction and cross-modal integration. For Qwen2.5-VL-3B, reasoning prompts structure latent knowledge, reduce uncertainty, and guide coherent intermediate steps. In contrast, InfiMed models have learned efficient, domain-specific strategies for direct medical answering, and enforcing explicit reasoning can disrupt these pathways, lowering performance on MedXpertQA-MM. Meanwhile, many other benchmarks are knowledge-driven, where answers can often be derived directly from visual information or domain expertise. For such tasks, step-by-step reasoning introduces redundant steps, increases hallucination risk, and interferes with the streamlined strategies of medically optimized models.

To further examine whether model size influences this phenomenon, we conducted additional experiments using larger model scales as well as different model families, including the LINGSHU (Xu et al., 2025) and HUATUO (Chen et al., 2024b) series. As shown in Figure 7 in the Appendix, even at the 7B and 32B scales, we continue to observe instances where explicit reasoning leads to performance degradation on several benchmarks. These results align with our primary conclusion that explicit reasoning is not universally advantageous for medical multimodal tasks. The degradation observed in larger models further suggests that introducing explicit reasoning can interfere with model performance on certain tasks.

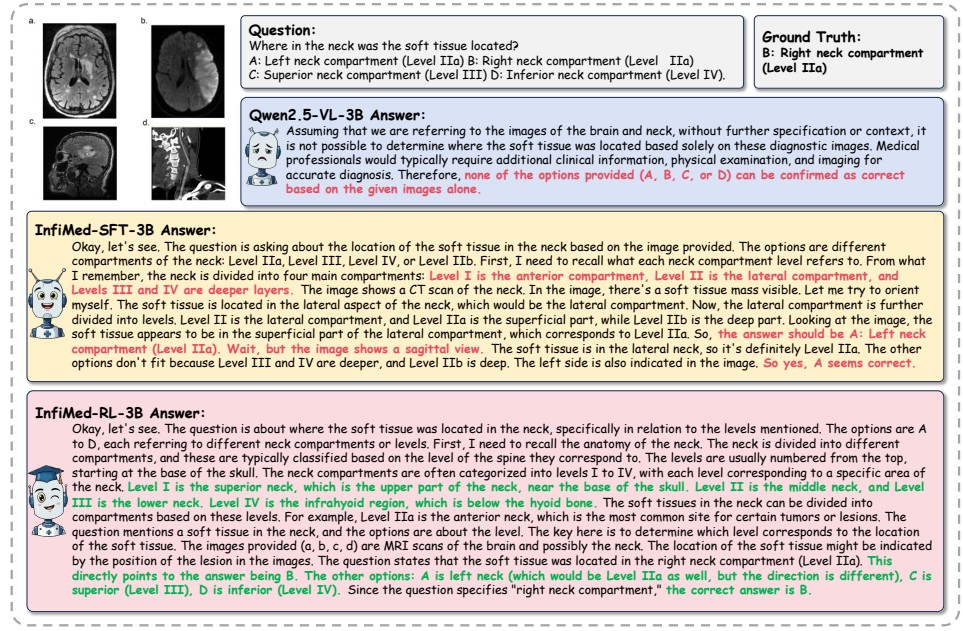

Figure 5: Case study on Qwen2.5-VL-3B, InfiMed-Series models in medical VQA. Red denotes errors or irrelevant content, whereas Green denotes correct or important information.

In summary, these results suggest that *the effectiveness of explicit reasoning depends on both task type and model.* It may benefit reasoning-intensive tasks and general-purpose models on medical tasks requiring light reasoning, but can hinder performance on recognition-oriented benchmarks where direct factual or visual knowledge suffices.

### 4.5 CASE STUDY (RQ4)

Beyond benchmark evaluations, we conduct a case study to examine the qualitative differences between the Qwen2.5-VL-3B-Instructw, reflective-pattern-injected InfiMed-SFT-3B, and the RLVR-optimized InfiMed-RL-3B, as illustrated in Figure 5. Our case study highlights distinct response behaviours across models. The general-purpose Qwen2.5-VL-3B adopts a conservative strategy, emphasizing its lack of sufficient medical knowledge and ultimately failing to produce a definitive answer. InfiMed-SFT-3B, by contrast, can generate a reasoning chain and reproduce a reflective pattern. However, despite this reflection, it still converges on the same incorrect answer. This suggests that SFT primarily teaches the model to mimic the form of reflection, yet falls short of enabling genuine understanding or effective application of reflective reasoning. InfiMed-RL-3B, on the other hand, demonstrates a more structured reasoning process. In addition to identifying the correct option, it actively explores and evaluates the other options, reflecting the impact of RLVR in pushing the model beyond memorized patterns toward deliberate and systematic reasoning. More case studies are presented in the Appendix A.6.

## 5 CONCLUSION

In this work, we introduce the InfiMed-Series models, including InfiMed-SFT-3B and InfiMed-RL-3B, a set of multimodal large language models (MLLMs) specialized for medical tasks. To address the scarcity and sparsity of multimodal medical data, we augmented the training sets with general multimodal and textual medical data and synthesized reflective-pattern-injected chain-of-thought data, enabling the models to acquire initial exploratory capabilities and providing a structured foundation for subsequent Reinforcement Learning with Verifiable Rewards (RLVR) training. Experimental results across diverse medical benchmarks, covering both reasoning-intensive and understanding-oriented tasks, show that the InfiMed-Series models achieve state-of-the-art accuracy among models with similar parameter counts and even surpass some larger models. Beyond performance gains, our analysis provides new insights into the behavior and potential of MLLMs in medical scenarios.

ETHICS STATEMENT

This work adheres to the ICLR Code of Ethics. Our study is purely empirical in nature, focused on advancing the field of medical multimodal large language models. We have exclusively used standard and publicly available medical datasets and open-source models, which were accessed and applied in strict accordance with their licenses.

We acknowledge the significant potential broader impacts and risks associated with the use of MLLMs in healthcare. This includes concerns related to patient safety, clinical accuracy, and the potential for misuse. Our work recognizes these challenges and aims to develop more powerful medical MLLMs that can contribute to society and human well-being. Importantly, our research does not introduce any new, unproven clinical applications. Instead, we focus on the ethical implications and underlying principles of utilizing MLLMs in healthcare settings, and we are committed to ensuring that future developments in this area are guided by a careful consideration of these ethical concerns.

REPRODUCIBILITY STATEMENT

To ensure the reproducibility of the results in this work, all models and synthetic datasets will be made publicly available. We also describe the data sources and detailed experimental setup in Section 4.1 and Appendix A.4.

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

# A APPENDIX

## A.1 THE USE OF LARGE LANGUAGE MODELS

We used a Large Language Model (LLM) **exclusively for editing purposes**, focusing on correcting grammar and typing errors. It's crucial to clarify that **the LLMs were not involved in the core aspects of the research**, including the development or revision of key ideas and experimental design.

## A.2 CONSTRUCTION OF REFLECTIVE-PATTERN-INJECTED COT

In this section, we present the detailed construction process of the reflective-pattern-injected CoT.

For multimodal datasets with sparse information (e.g., multiple-choice questions), each query is defined as $q = \{x, \mathcal{I}\}$, where $x$ denotes the textual task instruction and $\mathcal{I}$ represents one or more images. We first employ Qwen2.5-VL-32B (Bai et al., 2025) to generate 10 candidate responses $\{y_i\}_{i=1}^{10}$ for each query $q$. Through rejection sampling, we divide these into two subsets: $\{y_i^+\}_{i=1}^{m}$ and $\{y_i^-\}_{i=1}^{n}$, where $m + n = 10$.

For each response in $\{y_i^+\}_{i=1}^{m}$, we apply the following prompt to generate a score:

---

**Prompt for CoT Quality Evaluation**

You are a medical reasoning evaluator. Assess the following response based on these criteria:
**1. Clinical accuracy:** Correct incorporation of medical facts, clinical guidelines, and evidence-based practices. Accuracy, relevance, and appropriateness of clinical details.
**2. Logical reasoning:** Coherent reasoning process, logically leading to the answer, well-supported by clinical knowledge.
**3. Factual correctness:** All statements are factually correct and consistent with established medical knowledge.
**4. Completeness:** Thorough coverage of all necessary aspects without missing critical information.
**Question:** $\{q\}$
**Response:** $\{y_i^+\}$
Please evaluate the response on the above criteria and ONLY provide the `Dict` object with two keys:
{'score': integer between 1 and 10, 'justification': concise explanation of the score. }

---

After that, we compute the pass@10 for each query $q$, which corresponds to the number of correct responses among the 10 generated candidates, i.e., $m$. For $m \geq 6$, we directly select the $y_i^+$ with the highest score as the generated CoT. If multiple $y_i^+$ share the highest score, we randomly choose one.

For queries with $1 \leq m \leq 5$, we synthesize a reflective-pattern-injected CoT. Specifically, we first select one of the correct responses $y_i^+$ with the highest score and then randomly select one of the incorrect responses $y_i^-$. The reflective-pattern-injected CoT is subsequently synthesized through the following operation:

---

**Synthesis of the reflective-pattern-injected CoT**

$\{y_i^-\}$ Wait, perhaps we could consider it from a different perspective. Let's re-evaluate the problem step by step to ensure accuracy. $\{y_i^+\}$

---

Finally, we obtain CoT data enriched with reflective patterns through the integration of the aforementioned data, and we will release it once it is ready.

## A.3 REWARD FUNCTION

The other tasks' reward functions are as follows:

- **Mathematical Tasks:** For tasks involving mathematical expressions or numerical answers, $R_{\text{accuracy}}(o, \text{gt})$ is determined by a specialized verification function, denoted `math_verify`$(o_{\text{ans}}, \text{gt})$. This function evaluates the extracted answer from the output $o$, denoted $o_{\text{ans}}$, against the ground truth answer $gt$. The `math_verify` function is designed to handle nuances of mathematical evaluation, potentially allowing for symbolic equivalence or specified numerical tolerances. A successful verification yields a reward of 1; otherwise, 0.
- **Grounding Tasks:** For tasks where a model predicts a bounding box, we use the Intersection over Union (IoU) as the reward. This score measures the overlap between the predicted and ground-truth bounding boxes.

### A.4 DETAILS OF THE EXPERIMENTAL SETUP

**Training Datasets.** In the SFT stage, we use a total of **188K** training samples from three categories: (1) multimodal general data (LLaVA-OneVision (Li et al., 2024a)), (2) multimodal medical data (VQA-RAD (Lau et al., 2018), SLAKE (Liu et al., 2021), PathVQA (He et al., 2020), PMC-VQA (Zhang et al., 2023c), and our synthetic reflective-pattern-injected CoT), and (3) text-based medical data (ReasonMed (Sun et al., 2025), Medical-R1-Distill (Chen et al., 2024a), and Medical-o1-Reasoning (Chen et al., 2024a)). During the RLVR stage, we employ **36K** samples from multimodal general and medical datasets, including VQA-RAD, SLAKE, PathVQA, PMC-VQA, GMAI-Reasoning (Su et al., 2025), IconQA (Lu et al., 2021), ScienceQA (Lu et al., 2022a), TabMWP (Lu et al., 2022b), TQA (Zhou, 2025), and ViRFT_COCO (Liu et al., 2025e).

A detailed description of each dataset is provided as follows:

- LLaVA-OneVision (Li et al., 2024a): LLaVA-OneVision is a large-scale multimodal dataset comprising 4.8 million samples collected from diverse sources. It includes single-image, multi-image, and video modalities, and is specifically designed to train vision-language models for unified visual and textual understanding.
- VQA-RAD (Lau et al., 2018): VQA-RAD is a medical visual question answering dataset constructed for assessing multimodal understanding of radiology. It consists of radiological images paired with manually curated question-answer pairs authored by clinical experts. The dataset includes both open-ended and binary (yes/no) questions.
- SLAKE (Liu et al., 2021): SLAKE is a medical visual question answering dataset comprising 642 annotated radiological images spanning 39 anatomical structures and 12 disease categories. The dataset includes conditions such as various cancers (e.g., brain, liver, kidney, lung) and thoracic diseases (e.g., atelectasis, pleural effusion, pulmonary masses, and pneumothorax).
- PathVQA (He et al., 2020): PathVQA is a large-scale dataset developed for medical visual question answering tasks in the domain of pathology. It comprises 32,799 expert-annotated question-answer pairs spanning seven question categories, grounded in 4,998 high-resolution pathology images. The dataset includes both binary (yes/no) and open-ended questions.
- PMC-VQA (Zhang et al., 2023c) PMC-VQA is a large-scale medical visual question answering dataset designed to facilitate research on multimodal understanding in the medical domain. It comprises 227K VQA pairs grounded in 149K medical images, covering a wide range of imaging modalities and disease types.
- ReasonMed (Sun et al., 2025): ReasonMed is the largest open-source medical textual reasoning dataset containing 370K QA examples, which is distilled and filtered from three competitive large-language models (Qwen-2.5-72B, DeepSeek-R1-Distill-Llama-70B, and HuatuoGPT-o1-70B).
- Medical-R1-Distill (Chen et al., 2024a): Medical-R1-Distill-Data is an SFT dataset distilled from the DeepSeek-R1, constructed on verifiable medical questions from HuaTuoGPT-o1. It provides reasoning chains for medical problems, enabling the initialization and supervision of models' reasoning processes in the medical domain.
- Medical-o1-Reasoning (Chen et al., 2024a): medical-o1-reasoning-SFT is a SFT dataset focused on verifiable medical problems, where candidate solutions are generated by GPT-4o and validated by a medical verifier, providing high-quality reasoning chains and answers for training medical reasoning models.
- GMAI-Reasoning (Su et al., 2025): GMAI-Reasoning10K is a high-quality medical visual reasoning dataset comprising 10K curated multiple-choice questions constructed from 95 publicly available medical datasets spanning 12 imaging modalities (e.g., X-ray, CT, MRI). Each question is paired with standardized visual inputs and metadata, and generated using GPT-based prompting, following rigorous preprocessing and quality control procedures.

- IconQA (Lu et al., 2021): IconQA is a large-scale dataset containing 107,439 questions designed to assess abstract icon image understanding and visual language reasoning abilities.
- ScienceQA (Lu et al., 2022a): ScienceQA contains 21k multimodal questions, which align with California Common Core Content Standards, covering diverse science domains, many enriched with images, lectures, and explanations to support reasoning-oriented training.
- TabMWP (Lu et al., 2022b): Tabular Math Word Problems (TabMWP) is a multimodal dataset designed for training models to solve math word problems using both textual and tabular data. It contains 38,431 problems spanning elementary to high school levels, including both free-text and multiple-choice questions.
- TQA (Zhou, 2025): Textbook Question Answering (TQA) is a multimodal dataset designed for training models to answer questions using both textual and visual content from middle school science textbooks. Each sample provides a question, relevant textual context, and associated images, enabling models to learn to reason over multimodal inputs and generate accurate answers.
- ViRFT_COCO (Liu et al., 2025e): ViRFT_COCO is a vision-language dataset derived from COCO, containing around 6,000 samples. It aims to enhance models' ability to detect all instances of a given category within an image and output the corresponding bounding boxes with confidences under strict formatting constraints.

**Implementation Details.** Our InfiMed-Series models include InfiMed-SFT-3B and InfiMed-RL-3B.

- InfiMed-SFT-3B, which is built upon Qwen2.5-VL-3B (Bai et al., 2025), is trained using LLaMA-Factory (Zheng et al., 2024). We utilize 8 NVIDIA H800 GPUs. The vision tower and multimodal projector are frozen during training, while the language model remains fully trainable. We use a cosine learning rate scheduler with an initial learning rate of $5 \times 10^{-6}$, a warmup ratio of 0.1, and train for 5 epochs. The batch size is set to 4 per device. Furthermore, we set the maximum input resolution to 262,144 pixels for images, while text inputs are truncated to a maximum length of 4,096 tokens.
- InfiMed-RL-3B is built upon InfiMed-SFT-3B via EasyR1 (Sheng et al., 2024). For the RLVR reward function $R_{\text{total}}(o, \text{gt}) = w_{\text{format}} \cdot R_{\text{format}}(o) + w_{\text{acc}} \cdot R_{\text{accuracy}}(o, \text{gt})$, we set the weights $w_{\text{format}} = 0.1$ and $w_{\text{acc}} = 0.9$. All experiments were conducted using 16 NVIDIA H800 GPUs. For each phase, we used a learning rate of $1.0 \times 10^{-6}$, a batch size of 256 for training updates, a rollout batch size of 256, and generated 16 rollouts per sample during policy exploration.

**Evaluation Framework** To ensure consistency with prior work and a comprehensive, standardized evaluation, we adopt MedEvalKit (Xu et al., 2025), a systematic framework that integrates mainstream medical benchmarks and task types, supporting a range of question formats, including multiple-choice questions, open-ended questions, and closed-ended questions. We adopt the multimodal evaluation component of the framework, combining rule-based methods with the LLM-as-a-Judge strategy.

**Evaluation Benchmarks** We evaluate our InfiMed-Series models on seven widely used multimodal medical benchmarks, assessing both their reasoning ability and their understanding of medical knowledge. The detailed description of the benchmarks is as follows:

- MMMU (Yue et al., 2024): MMMU is a benchmark designed to assess the capabilities of multimodal models on large-scale, multidisciplinary tasks. It comprises 11.5K meticulously curated multimodal questions drawn from university exams, quizzes, and textbooks, covering six core disciplines, including Health & Medicine. The Health & Medicine includes 1,752 test questions—accounting for 17% of the entire benchmark—and is further subdivided into five specialized domains: Basic Medical Science, Clinical Medicine, Diagnostics and Laboratory Medicine, Pharmacy, and Public Health.
- VQA-RAD (Lau et al., 2018): VQA-RAD is a dataset consisting of question–answer pairs grounded in radiological medical images, intended for training and evaluating medical visual question answering systems. It includes both open-ended questions and binary yes/no questions. In total, the dataset comprises 2,248 QA pairs linked to 315 medical images, with all annotations manually curated by a team of clinicians to ensure clinical relevance and accuracy.
- SLAKE (Liu et al., 2021): SLAKE is a bilingual (Chinese-English) dataset specifically designed for medical visual question answering systems. It consists of 642 medical images paired with 14,028 question-answer instances.
- PathVQA (He et al., 2020): PathVQA is designed for visual question answering in the field of pathology. It comprises 4,998 pathology images collected from two pathology textbooks and the PEIR digital library, accompanied by a total of 32,799 question-answer pairs.

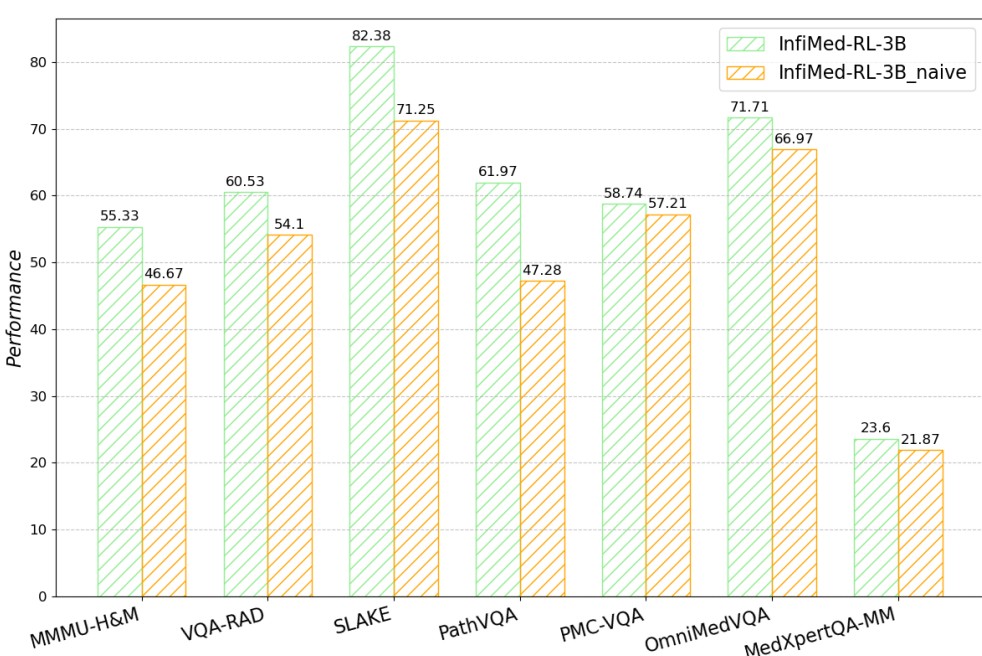

Figure 6: Performance comparison of InfiMed-RL-3B and InfiMed-RL-3B_naive on medical benchmarks. InfiMed-RL-3B_naive denotes directly utilizing RLVR upon Qwen2.5-VL-3B.

- PMC-VQA (Zhang et al., 2023c): PMC-VQA is a large-scale multimodal dataset constructed for medical visual question answering. It contains 227,000 VQA questions grounded in 149,000 medical images spanning a wide range of imaging modalities and disease types.
- OmniMedVQA (Hu et al., 2024): OmniMedVQA is a large-scale and comprehensive visual question answering benchmark tailored specifically for the medical domain. It aggregates data from 73 distinct medical datasets, comprising 118,010 images and 127,995 question-answer pairs. The benchmark encompasses 12 different medical imaging modalities and covers more than 20 anatomical regions of the human body.
- MedXpertQA (Zuo et al., 2025): MedXpertQA is a benchmark specifically designed to evaluate professional medical knowledge. It comprises 4,460 questions spanning 17 medical specialties and 11 organ systems. In our experiments, we utilize only the multimodal subset of the dataset.

## A.5 ABLATION STUDY ON RLVR

In this subsection, we present ablation studies related to RLVR, where we explore training from two different starting points: one from the Qwen2.5-VL-3B model (Bai et al., 2025), and the other from our InfiMed-SFT-3B. The results are presented in Figure 6.

Notably, in tasks requiring the integration of large amounts of domain-specific information, such as VQA-RAD, PathVQA, and SLAKE, the InfiMed-RL-3B_naive model significantly underperforms compared to InfiMed-RL-3B. This suggests that directly applying RLVR to Qwen2.5-VL-3B without incorporating domain-specific data during the SFT phase can lead to much lower performance, especially on tasks that require understanding and memorizing domain-specific knowledge. This underscores the importance of the cold start phase, where injecting relevant, knowledge-rich data during SFT is essential to building a solid foundation for the subsequent RLVR phase.

Additionally, in MMMU-H&M, InfiMed-RL-3B achieves a significantly higher score of 55.33, compared to 46.67 for InfiMed-RL-3B_naive. Given that this benchmark demands both reasoning and comprehensive multimodal understanding (general and medical), this highlights the critical role of the SFT phase in helping the model integrate complex information effectively. The results on PMC-VQA and MedXpertQA-MM further demonstrate that prior domain-specific fine-tuning improves RLVR training outcomes.

Table 3: **Ablation study examining data composition during the RLVR stage.** $\Delta|\text{Data}|$ denotes the changes of the training dataset.

| Model | $\Delta|\text{Data}|$ | Accuracy (%) | | | | | | | |
|---|---|---|---|---|---|---|---|---|---|
| | | MMMU-H&M | VQA-RAD | SLAKE | PathVQA | PMC-VQA | OMVQA | MedXQA | Avg. |
| *Base Model* | | | | | | | | | |
| Qwen2.5VL-3B | - | 51.3 | 56.8 | 63.2 | 37.1 | 50.6 | 64.5 | 20.7 | 49.2 |
| *Ablation Study in RLVR Stage* | | | | | | | | | |
| InfiMed-RL-3B | - | 55.3 | 60.5 | 82.4 | 62.0 | 58.7 | 71.7 | 23.6 | 59.2 |
| InfiMed-RL-3B+medical_mm | +16K | 56.0 | 60.9 | 82.2 | 61.5 | 58.4 | 70.0 | 23.6 | 58.9 |
| InfiMed-RL-3B-w/o-general | −10K | 53.3 | 60.7 | 81.9 | 61.6 | 58.3 | 70.0 | 23.6 | 58.4 |

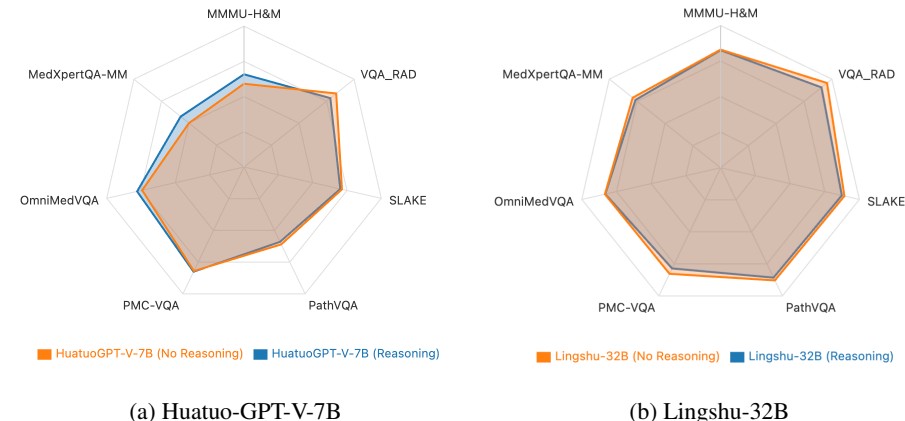

(a) Huatuo-GPT-V-7B                    (b) Lingshu-32B

Figure 7: Comparison of direct-answer and reasoning-based prompts on medical benchmarks with larger models.

We further explored whether enlarging the RLVR dataset could lead to additional performance gains. To this end, we constructed a larger RLVR set by incorporating 16K additional medical questions sourced from SLAKE (5K), PathVQA (9K), and VQA-RAD (2K). These questions were selected by prioritizing cases in which InfiMed-SFT-3B produced the fewest correct responses across 10 attempts. The experimental results in the Table 3 show that simply increasing the number of RLVR training questions does not consistently yield performance improvements. This finding suggests that, although additional RLVR data may still provide incremental benefits, the marginal gains are limited given the substantial effort required to curate high-quality medical multimodal RLVR data.

## A.6 CASE STUDIES ON QWEN2.5-VL-3B, INFIMED-SFT-3B, AND INFIMED-RL-3B

In this section, we present additional case studies to illustrate the distinct responses of Qwen2.5-VL-3B, InfiMed-SFT-3B, and InfiMed-RL-3B. In summary, our analysis reveals that the InfiMed-SFT-3B model already obtains self-reflective ability but is prone to delivering redundant responses, whereas the InfiMed-RL-3B model minimizes verbosity while ensuring accuracy in its answers.

This change is consistent with the characteristics of the GRPO optimization mechanism. During GRPO training, the gradient update for each rollout response is normalized by its length, which amplifies the learning signal for shorter and correct responses. Consequently, the reinforcement learning stage naturally encourages the model to prefer concise and accurate answers rather than extended reflective chains. The reflective supervision used during SFT primarily serves to enrich the model's reasoning search space for RLVR, rather than to enforce reflective verbosity in the final model. After RLVR, the model shifts from imitating reflection-style formats to producing more efficient outputs, which explains the observed reduction in explicit reflective content.

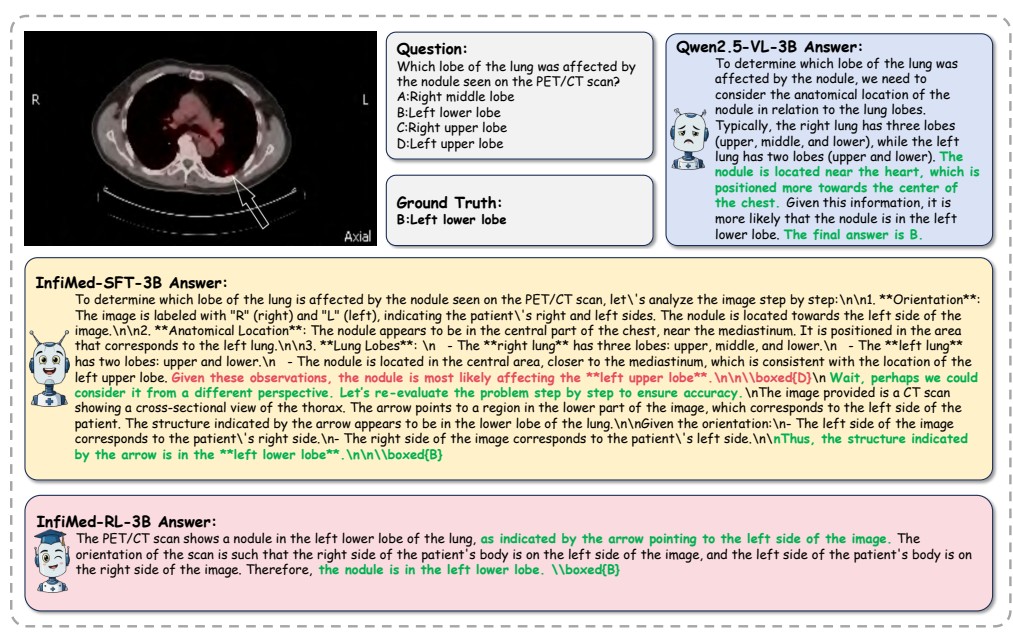

Figure 8: Case study on Qwen2.5-VL-3B, InfiMed-Series models in medical VQA. Red denotes errors or irrelevant content, whereas Green denotes correct or important information. All of the models answered correctly; InfiMed-SFT-3B answered correctly after reflection.

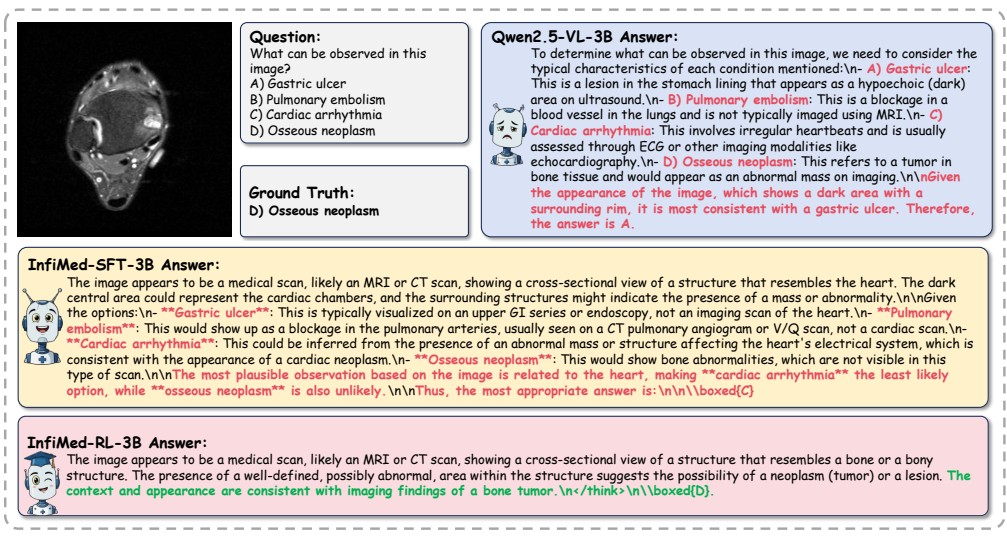

Figure 9: Case study on Qwen2.5-VL-3B, InfiMed-Series models in medical VQA. Red denotes errors or irrelevant content, whereas Green denotes correct or important information. Only InfiMed-RL-3B answered correctly, providing a direct and accurate response by correctly identifying the key visual information and selecting the correct answer with a concise and focused explanation.

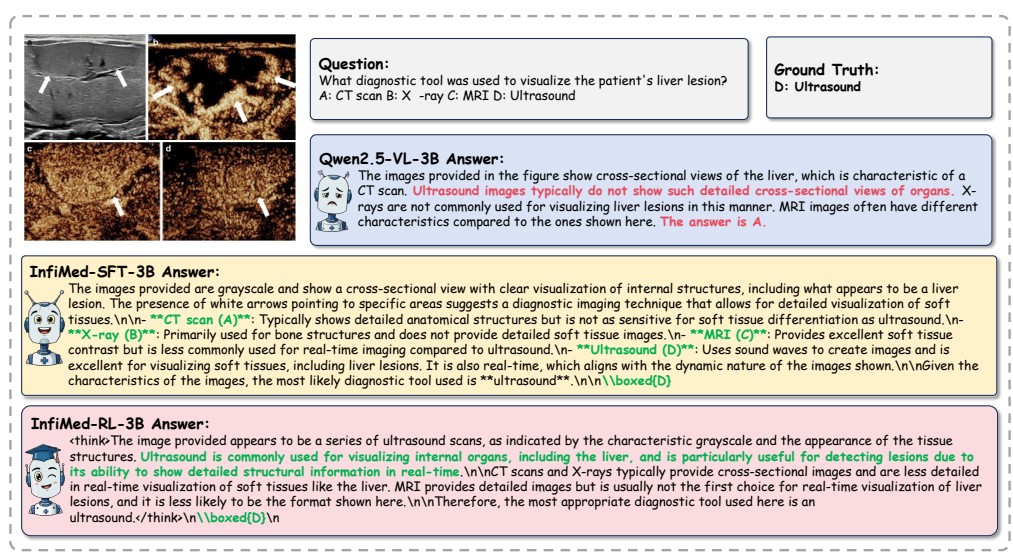

Figure 10: Case study on Qwen2.5-VL-3B, InfiMed-Series models in medical VQA. Red denotes errors or irrelevant content, whereas Green denotes correct or important information. Qwen2.5-VL-3B's response is incorrect because it fundamentally fails to recognize the visual characteristics of an ultrasound scan, leading to a flawed conclusion. InfiMed-SFT-3B provides a detailed, step-by-step reasoning process, while InfiMed-RL-3B offers a more direct and accurate answer, showcasing its improved ability to instantly recognize diagnostic imaging types.

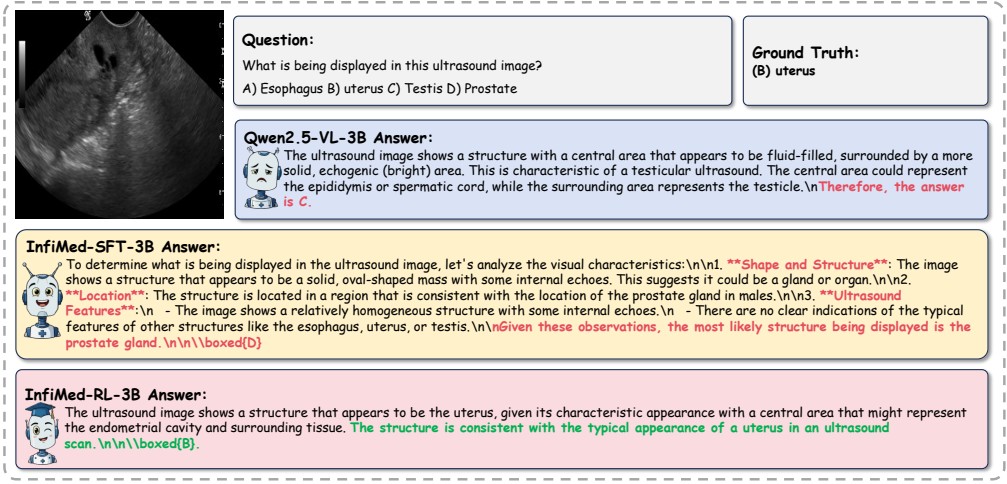

Figure 11: Case study on Qwen2.5-VL-3B, InfiMed-Series models in medical VQA. Red denotes errors or irrelevant content, whereas Green denotes correct or important information. Incorrectly identifies the image and chooses the wrong answer. InfiMed-SFT-3B provides a detailed analysis that correctly rules out most options but ultimately guesses the wrong answer. InfiMed-RL-3B is the only model that correctly identifies the organ shown, providing the most direct and accurate response.

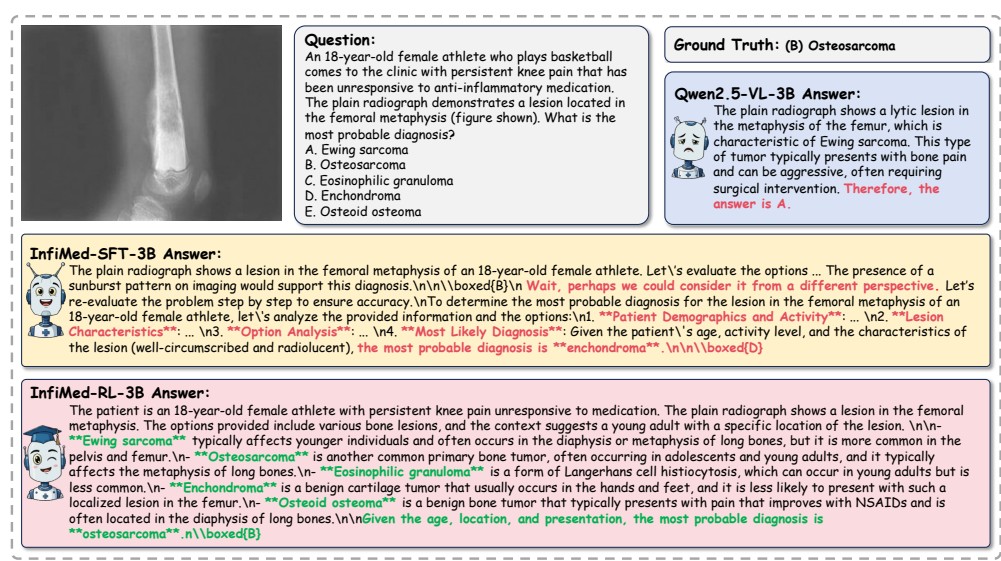

Figure 12: Case study on Qwen2.5-VL-3B, InfiMed-Series models in medical VQA. Red denotes errors or irrelevant content, whereas Green denotes correct or important information. Qwen2.5-VL-3B incorrectly identifies the lesion. InfiMed-SFT-3B initially makes a correct deduction but then overthinks the problem, leading to a self-correction that results in the wrong answer. InfiMed-RL-3B provides a detailed and accurate analysis.

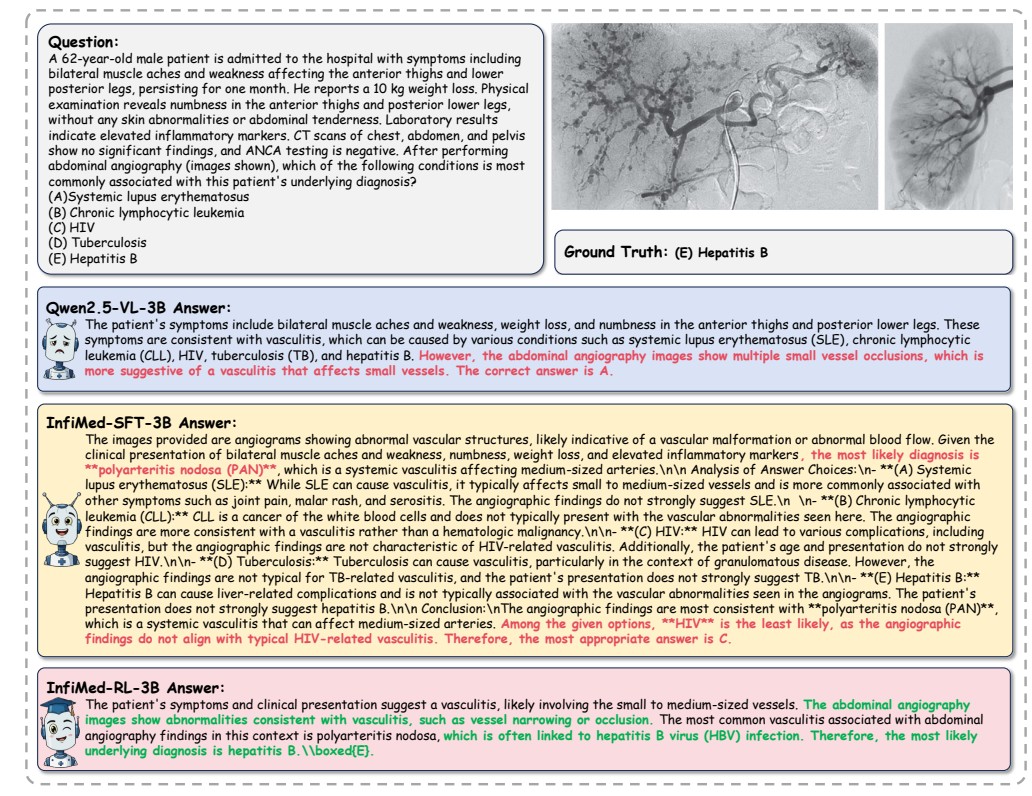

Figure 13: Case study on Qwen2.5-VL-3B, InfiMed-Series models in medical VQA. Red denotes errors or irrelevant content, whereas Green denotes correct or important information. Only InfiMed-RL-3B successfully synthesizes the angiographic findings and patient symptoms to provide the correct diagnosis.

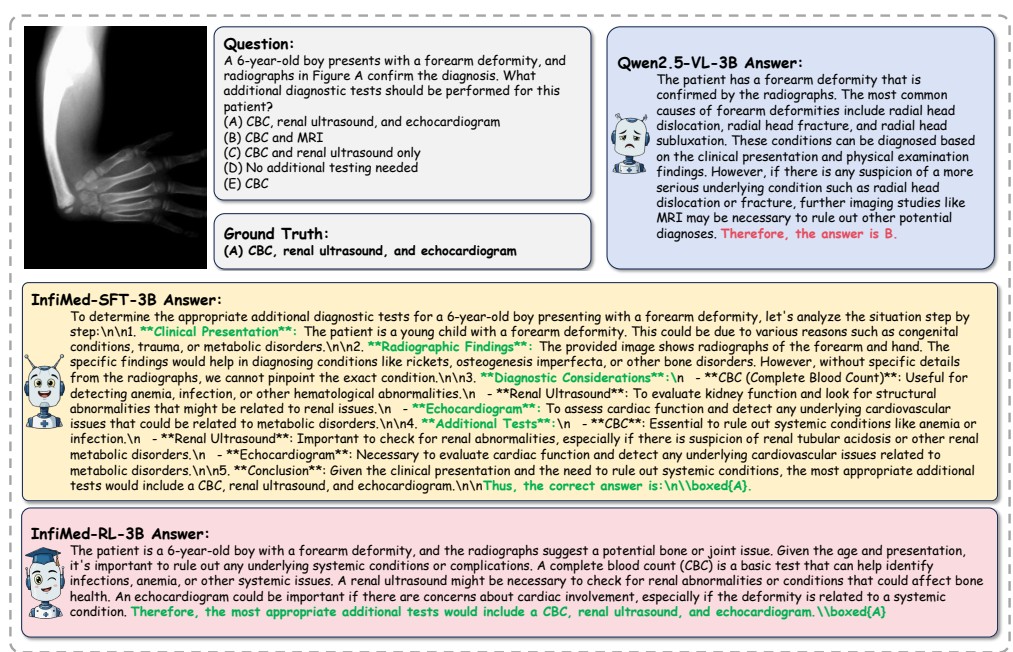

Figure 14: Case study on Qwen2.5-VL-3B, InfiMed-Series models in medical VQA. Red denotes errors or irrelevant content, whereas Green denotes correct or important information. Qwen2.5-VL-3B incorrectly identifies the necessary tests. Both InfiMed-SFT-3B and InfiMed-RL-3B correctly solve this case, but InfiMed-RL-3B provides a more direct and efficient answer.

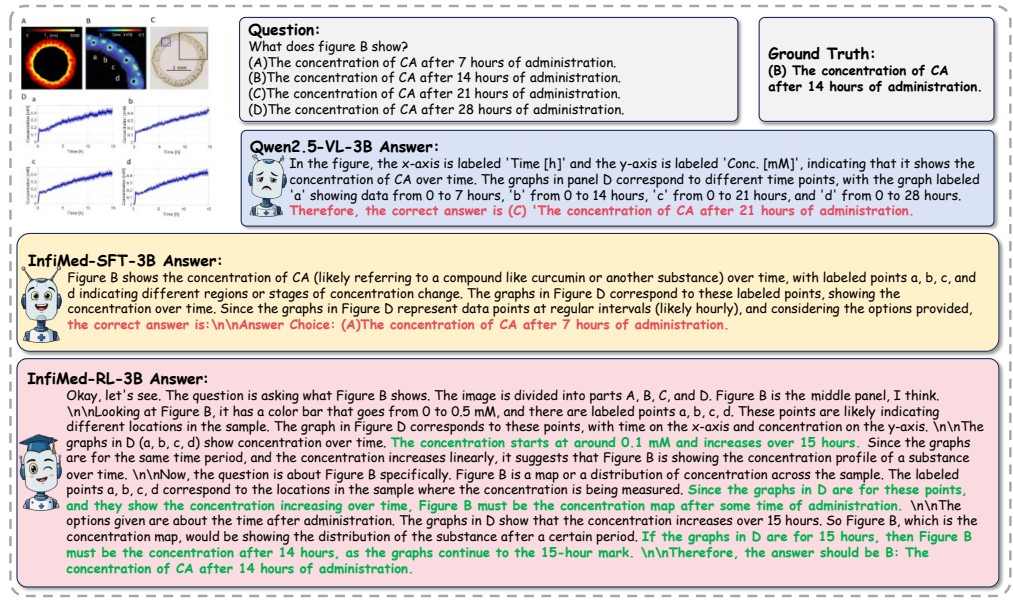

Figure 15: Case study on Qwen2.5-VL-3B, InfiMed-Series models in medical-related VQA (MMMU-H&M). Red denotes errors or irrelevant content, whereas Green denotes correct or important information. Qwen2.5-VL-3B incorrectly identifies the relevant data series. InfiMed-SFT-3B fails to correctly interpret the graphs and provides an incorrect analysis. Only InfiMed-RL-3B correctly synthesizes the information from both the concentration map and the graphs to provide the right answer.

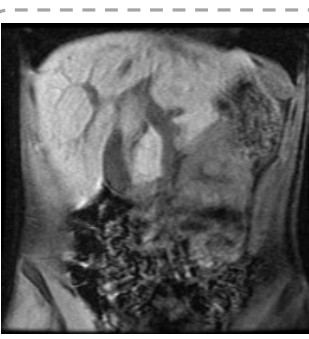

**Question:**
Does this image show any variation from normal?
A. It's difficult to determine if there is an abnormality based on this image.
B. There is a possibility of abnormality, but further tests are needed for confirmation.
C. I'm not sure, but there might be a slight abnormality.
D. No, It's normal.

**Ground Truth:**
D. No, It's normal.

**Qwen2.5-VL-3B Answer:**
To determine if there is an abnormality in the image, we need to consider the typical appearance of normal tissues and structures. The image shows a cross-sectional view of what appears to be a body part, possibly the neck or throat area, with various structures visible.\n\n1. **Identify Normal Structures**: In a normal CT scan of the neck or throat, you would expect to see structures such as the thyroid gland, larynx, trachea, and other soft tissues. These structures should have specific shapes and densities that are characteristic of their normal state.\n\n2. **Analyze Abnormalities**: If there are any deviations from these expected shapes or densities, it could indicate an abnormality. For example, an enlarged thyroid gland, a mass in the larynx, or changes in the tracheal wall could suggest pathology.\n\n3. **Consider the Image**: Based on the provided image, there are no obvious signs of gross abnormalities like masses, calcifications, or unusual shapes that would typically indicate pathology. However, without a detailed analysis by a radiologist who specializes in interpreting such images, it is challenging to definitively say whether there is an abnormality.\n\n Given the options:\n- Option A suggests it's difficult to determine if there is an abnormality based on this image alone.\n- Option B implies there is a possibility of abnormality but requires further testing.\n- Option C suggests there might be a slight abnormality, which is less definitive than option A.\n- Option D states it's normal, which is too definitive given the lack of clear indicators of abnormality.\n\n Considering all factors, the most appropriate answer is: A

**InfiMed-SFT-3B Answer:**
To determine if the image shows any variation from normal, let's analyze the visual characteristics:\n\n1. **Overall Structure**: The image appears to show a cross-sectional view of a structure, possibly a part of the body like the brain or spinal cord. The symmetry and structure seem intact.\n\n2. **Anatomical Features**: There are no obvious structural abnormalities such as holes, irregularities, or unusual shapes that would suggest a pathological condition.\n\n3. **Color and Texture**: The colors and textures appear consistent with what would be expected in a normal anatomical structure. There are no areas that stand out as abnormal in terms of color or texture.\n\n4. **Comparison to Normal Expectations**: Without a reference image or additional context, it is challenging to definitively say whether there is an abnormality. However, the image does not show any clear signs of pathology.\n\nGiven these observations, the most appropriate answer is:\n\n\\boxed{A}. Wait, perhaps we could consider it from a different perspective. Let's re-evaluate the problem step by step to ensure accuracy. \nTo determine if the image shows any variation from normal, let's analyze the provided MRI scan step by step:\n\n1. **General Observations**:\n - The image appears to be an MRI scan of the abdomen or pelvis.\n - There are structures visible, including what looks like the spine, surrounding tissues, and possibly organs.\n\n2. **Specific Features**:\n - The central area appears to have a relatively uniform structure, which is typical for normal tissue.\n - There are no obvious signs of abnormal masses, lesions, or distortions in the immediate vicinity of the spine and surrounding structures.\n\n3. **Comparison to Normal Anatomy**:\n - Normal anatomy in this region would typically show well-defined structures without significant abnormalities.\n - The image does not show any clear signs of pathology such as tumors, inflammation, or other anomalies.\n\n4. **Conclusion**:\n - Based on the uniform appearance and lack of visible abnormalities, the image does not show any variation from normal.\n\nTherefore, the most appropriate answer is:\n\n\\boxed{D}

**InfiMed-RL-3B Answer:**
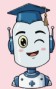
The image appears to be a medical scan, likely an MRI or CT scan, showing a cross-sectional view of the body. The presence of contrast and the structure suggest it could be a diagnostic image. However, without specific medical training or additional context, it's challenging to definitively identify any abnormalities. The appearance could be normal, but the presence of contrast and the complexity of the image might indicate the presence of some structural or physiological changes.\n\\boxed{D}

Figure 16: Case study on Qwen2.5-VL-3B, InfiMed-Series models in medical VQA. Red denotes errors or irrelevant content, whereas Green denotes correct or important information. Qwen2.5-VL-3B's response is incorrect, while InfiMed-SFT-3B correctly answers after a detailed analysis and reflection, and InfiMed-RL-3B provides the most direct and accurate correct answer.

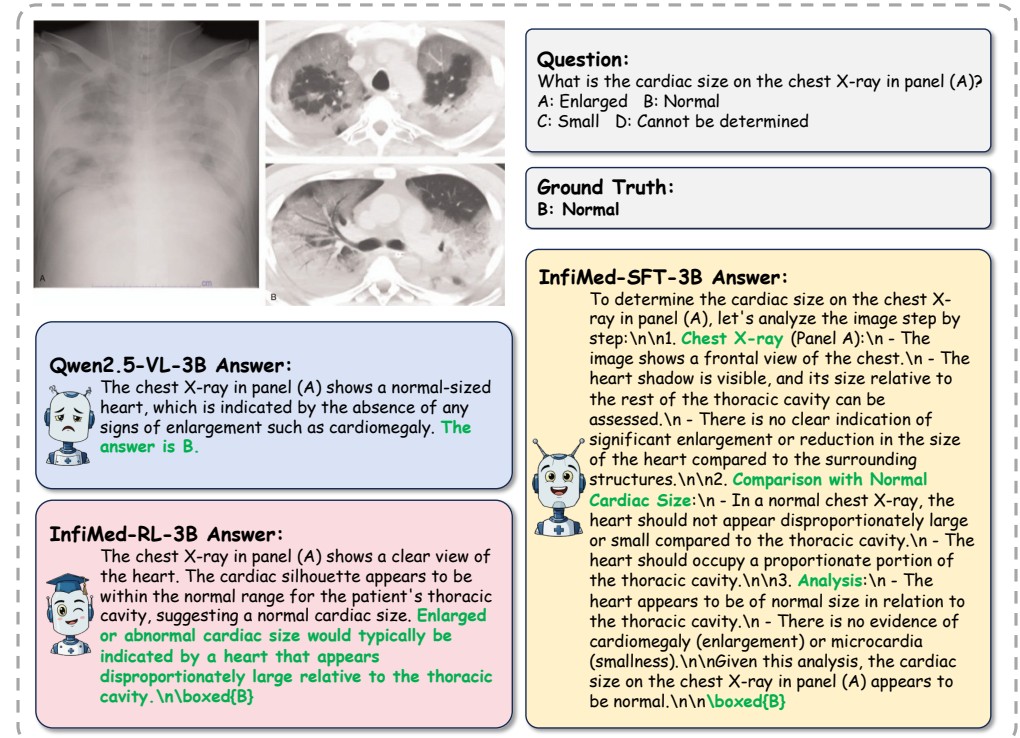

Figure 17: Case study on Qwen2.5-VL-3B, InfiMed-Series models in medical VQA. Red denotes errors or irrelevant content, whereas Green denotes correct or important information. All three models correctly identify the cardiac size as normal, with Qwen2.5-VL-3B providing a concise answer, InfiMed-SFT-3B offering a detailed, step-by-step analysis, and InfiMed-RL-3B giving a direct and well-reasoned response.

## A.7 LIMITATIONS AND FUTURE WORKS

Although our InfiMed-Series models achieve state-of-the-art (SOTA) performance among MLLMs with a similar number of parameters, they even outperform some MLLMs with larger parameter counts. However, it is undeniable that open-source medical multimodal data often exhibit low quality, including poor image resolution, non-uniform distribution of modalities, and errors introduced during model synthesis. Consequently, some of the results may lack full confidence, and the models' performance on more complex medical downstream tasks remains to be thoroughly explored. Moreover, how to develop reasoning steps that can be more efficient and accurate in the medical field is a critical issue that needs further study.

Additionally, we want to clarify that our training datasets are sourced exclusively from publicly available datasets, ensuring that no private or sensitive data is involved.

