# OpenReview forum: "InfiMed: Low-Resource Medical MLLMs with Advancing Understanding and Reasoning"
_ICLR.cc/2026/Conference — ICLR 2026 Conference Withdrawn Submission_

### Official Review · Reviewer_pyBe · 2025-10-29

**Soundness:** 3
**Presentation:** 3
**Contribution:** 2
**Rating:** 4
**Confidence:** 3

**Summary:**

This paper addresses the limited reasoning ability of multimodal medical large language models (MLLMs) in information-sparse scenarios by proposing a systematic solution. Through reflective-pattern-injected SFT and group-relative advantage-optimized RLVR, the model learns to self-evaluate, correct errors, and explore multi-step reasoning paths. Leveraging both general and medical multimodal data under low-resource conditions, the InfiMed series achieves strong performance across seven medical benchmarks, particularly excelling in reasoning-intensive tasks, with ablation studies and case analyses validating the effectiveness of the approach.

**Strengths:**

(1) The method is well designed. The paper integrates reflective Chain-of-Thought (CoT) injection with the RLVR framework to enhance the model’s self-evaluation and multi-step reasoning capabilities.
(2) The experiments cover multiple medical benchmark datasets, and the proposed method is validated through ablation studies and case analyses, confirming its effectiveness.

**Weaknesses:**

1. This study aims to enhance the model’s reasoning and understanding abilities. However, ablation results show that general multimodal data improve performance more than the reflective CoT, suggesting a potential inconsistency with the study’s focus.
2. The methodology and figures are unclear. It is recommended that the authors describe the inference trajectory of the general CoT to help readers understand the model’s reasoning mechanism. Additionally, they should further clarify why multimodal medical data and textual medical data are displayed in parallel in Figure 2, considering that text is already part of the multimodal data.
3. The related work section is incomplete. It lacks a discussion of the application of CoT in MLLMs, which limits the reader’s understanding of the technical positioning and novelty of this study. It is recommended that the authors include this discussion in the related work section to provide a complete technical background and highlight the novelty of the study.
﻿4. The experimental section omits ablation studies under conditions without general CoT data, making it difficult to evaluate the independent contribution of the method. It is recommended that the authors supplement ablation experiments under conditions lacking general CoT data to verify its independent contribution.﻿
5. In addition, there are minor issues, such as incorrect variable notations, which further undermine the overall clarity and rigor of the manuscript. It is recommended that the authors carefully check the details in the paper, such as the notation error in line 233 (“and c denotes the reward”), to ensure consistency and accuracy in expressions.

**Questions:**

Please refer to the Weaknesses.

---

> ### Author Response · Authors · 2025-11-24
>
> **Q1:** This study aims to enhance the model’s reasoning and understanding abilities. However, ablation results show that general multimodal data improve performance more than the reflective CoT, suggesting a potential inconsistency with the study’s focus.
> **Ans:** Thanks for the reviewer’s comment. Our study focuses on how to build a strong medical multimodal foundation model under the constraint of limited open-source medical data. In this process, we find that general multimodal data and medical textual data provide substantial performance gains on medical multimodal benchmarks, which is one of the key empirical observations of our work. The role of reflective CoT is fundamentally different from that of general data. Since the RLVR stage relies on the model’s ability to actively explore better reasoning paths, we incorporate reflective CoT during SFT primarily to expand the exploration space for RLVR, rather than to directly improve SFT performance. And our ablation study shows that reflective CoT still remains a meaningful component of the training pipeline.
>
> **Q2:** The methodology and figures are unclear.
> **Ans:** Thanks for the reviewer’s comment. To address the reviewer’s suggestion, we will add a brief introduction to the concept of general CoT in our paper. *_General CoT refers to chain-of-thought style reasoning traces constructed from instruction data. These traces typically include multi-step natural language explanations that guide the model through intermediate reasoning steps before producing the final answer, and they are used to strengthen the model’s ability to follow structured reasoning patterns during SFT_*.
> Regarding Figure 2, we separate multimodal medical data and textual medical data because they represent different input modalities in our setting. Specifically, multimodal medical data refers to samples that contain both images and text, whereas textual medical data contains only text without images. For this reason, we display them in parallel to clearly distinguish their roles and contributions in the training pipeline.
>
> **Q3:** The related work section is incomplete. It lacks a discussion of the application of CoT in MLLMs.
> **Ans:** Thanks for the comment. To address this suggestion and more clearly position our work, we will extend the related work section by introducing the concept of general CoT (as mentioned in Q2) and discussing its applications in MLLMs. The additional content we plan to include is as follows:
> *_General CoT refers to chain-of-thought style reasoning traces constructed from instruction data. These traces contain multi-step natural language explanations that guide the model through intermediate reasoning steps before producing the final answer, enabling the model to learn structured reasoning patterns during SFT. This allows the model to better select the most appropriate response among diverse candidate trajectories during the RLVR stage after it has learned how to solve tasks in SFT [1-3].
> However, medical tasks are often highly standardized and fact-driven, with very few acceptable reasoning paths. Medical reasoning typically requires strict adherence to clinical knowledge, leaving little room for varied intermediate steps.
> As a result, the exploration space available to RLVR is much narrower than in open-domain settings, making it more difficult for the model to discover diverse or improved reasoning trajectories during RLVR [4]. To expand the search space available to the model during RLVR, we introduce reflective-pattern–injected CoT data in the SFT stage. This equips the model with preliminary reflective and self-correction abilities, enabling RLVR to operate over richer reasoning trajectories._*
>
> [1] Deng, Y., Bansal, H., Yin, F., Peng, N., Wang, W., & Chang, K. W. (2025). Openvlthinker: Complex vision-language reasoning via iterative sft-rl cycles. In The Thirty-ninth Annual Conference on Neural Information Processing Systems.
> [2] Yang, Y., He, X., Pan, H., Jiang, X., Deng, Y., Yang, X., et al. (2025). R1-onevision: Advancing generalized multimodal reasoning through cross-modal formalization. arXiv preprint arXiv:2503.10615.
> [3] Ni M, Yang Z, Li L, et al. Point-rft: Improving multimodal reasoning with visually grounded reinforcement finetuning[J]. arXiv preprint arXiv:2505.19702, 2025.
> [4] Xu, W., Chan, H. P., Li, L., Aljunied, M., Yuan, R., Wang, J., et al. (2025). Lingshu: A Generalist Foundation Model for Unified Multimodal Medical Understanding and Reasoning. arXiv preprint arXiv:2506.07044.

---

> > ### Author Response · Authors · 2025-11-24
> >
> > **Q4:** The experimental section omits ablation studies under conditions without general CoT data.
> > **Ans:** Thanks for the comment. We would like to declare that CoT in the SFT stage follows standard practice in both LLMs and MLLMs [1-4], as it provides structured reasoning demonstrations and injects useful task knowledge that prepares the model for RLVR. We aim to build a strong medical multimodal foundation model under limited data conditions and to explore characteristics unique to the medical multimodal scenarios.
> > To directly address the reviewer’s concern, we have added ablations discussing the contribution of general CoT. Despite removing reflective CoT entirely (results are shown in  Section 4.3), we also replace reflective CoT with the same amount of general CoT, which is shown below. The superior results of adding reflective CoT indicate that the improvement does not come simply from adding CoT, but from the medical-reflective patterns contained in reflective CoT. These experiments verify the independent effectiveness of reflective CoT in our training pipeline.
> > | Model| MMMU-H&M | VQA-RAD |SLAKE | PathVQA | PMC-VQA | OmniMedVQA | MedXpertQA-MM | Avg.|
> > |-|-|-|-|-|-|-|-|-|
> > | InfiMed-SFT-w-generalCoT| 50.67|60.53| 81.94 |57.67| 52.78| 66.36|23.4|56.19|
> > | InfiMed-SFT-3B| 54.7|58.1|82|60.6|53.2|67| 23.5| 57.1 |
> >
> > **Q5:** Some typos.
> > **Ans:** Thanks for the comments. We will carefully check and correct the typos in the manuscript.

---

> > > ### Comment · Reviewer_pyBe · 2025-11-26
> > >
> > > According to the comments of other reviewers, I decided to keep the score.

---

### Official Review · Reviewer_GiWs · 2025-10-29

**Soundness:** 3
**Presentation:** 4
**Contribution:** 3
**Rating:** 4
**Confidence:** 4

**Summary:**

This paper proposes InfiMed for low-resource training for medical multimodal large language models. It uses a stronger language model to sample and filter reasoning traces on some general domain and medical datasets and use supervised fine-tuning to inject these capability to their model. A stage of reinforcement learning with verifiable rewards follows to further enhance the reasoning ability of the model. The authors also conducted many analysis on the relationship between the design of training data / training pipeline and the model performance, showing that reasoning is not always helpful for some tasks.

**Strengths:**

1. The paper is well written with illustrative figures and good explanations.
2. The results of the model are strong and can often surpass larger-scale models.
3. The analysis part of this paper is useful for future practitioners. It reveals the relationship between the training data and model performance on different benchmarks. Some conclusions are interesting, e.g. reasoning is not always helpful. Will be better if experiments can be done on larger scale or more recent models.

**Weaknesses:**

1. The methods in this paper are well known.  The reasoning construction, the training pipeline, i.e. SFT followed by RLVR, and the rewarding functions are used by many concurrent papers, and it is hard to distinguish this paper.
2. The conclusions are drawn from a 3B model.  As revealed by previous work (e.g. DeekSeek-R1 series), the reasoning capability is best incentivized only on large models.  Some conclusions of this paper demonstrate that reasoning might hurt the model performance on a few datasets, but it remains open question if using larger model can fix this.

**Questions:**

Suggestion on writings: The fonts on the figures are not professional. They are good for poster / slides but not ideal for paper presentation. Should have used serif to fit it into the ICLR formats (Times New Roman).

---

> ### Author Response · Authors · 2025-11-24
>
> **Q1:** The reasoning construction, SFT to RLVR training pipeline, and reward functions are widely used in concurrent works, making it difficult to identify the paper’s unique contributions.
> **Ans:** Thanks for the reviewer’s comment. We acknowledge that the overall training pipeline, including SFT, RLVR, and the general framework for post-training, has been adopted by several concurrent works. However, the focus of our paper is not on proposing a new training paradigm, but on demonstrating that a 3B-scale medical multimodal foundation model can achieve SOTA performance with limited medical data, and on uncovering domain-specific findings that are unique to the medical multimodal setting. In particular, we show that (1) incorporating general multimodal data together with medical textual data can effectively improve performance on medical multimodal tasks, (2) introducing reflective data during SFT provides a broader search space for the RL stage, allowing the model to explore more reasoning trajectories, and (3) explicit reasoning does not always provide benefits in medical tasks, which differs from observations commonly reported in general-domain settings. We believe these insights are valuable for the development of future medical multimodal foundation models and help differentiate our work from existing methods.
>
> **Q2:** Some conclusions of this paper demonstrate that reasoning might hurt the performance of the model on a few datasets, but it remains an open question if using a larger model can fix this.
> **Ans:** Thanks for the reviewer’s insightful comment. This is indeed an important question. To further examine whether model size affects the conclusion, we conducted additional experiments using both larger model sizes and different model families (the LINGSHU [1] series and the HUATUO [2] series). The results, summarized in the table, show that even at the 7B and 32B scales, we still observe cases where explicit reasoning leads to performance degradation on certain datasets. These findings are consistent with our original conclusion that explicit reasoning is not universally beneficial in medical multimodal tasks.
> | Model                     | MMMU-H&M | VQA-RAD | SLAKE  | PathVQA | PMC-VQA | OmniMedVQA | MedXpertQA-MM |
> |---------------------------|----------|---------|--------|---------|---------|------------|----------------|
> | HuatuoGPT-7B-w/o reasoning | 47.3     | 67      | 67.8   | 48      | 53.3    | 74.2       | 21.6           |
> | HuatuoGPT-7B-w-reasoning     | 52.67    | 62.75   | 66.62  | 44.25   | 53.66   | 77.86      | 23             |
> | Lingshu-32B-w/o reasoning  | 62.3     | 76.5    | 89.2   | 65.9    | 57.9    | 83.4       | 30.9           |
> | Lingshu-32B-w-reasoning  | 63.3       | 72.51   | 87.34  | 64.28   | 54.98   | 83.08      | 30.6           |
>
> **Q3:** Suggestion on writings.
> **Ans:** Thanks for the comments, and we will update the figures to better align with the ICLR formatting requirements.
>
> [1] Xu, W., Chan, H. P., Li, L., Aljunied, M., Yuan, R., Wang, J., et al. (2025). Lingshu: A Generalist Foundation Model for Unified Multimodal Medical Understanding and Reasoning. arXiv preprint arXiv:2506.07044.
> [2] Chen, J., Gui, C., Ouyang, R., Gao, A., Chen, S., Chen, G. H., et al. (2024). Huatuogpt-vision, towards injecting medical visual knowledge into multimodal llms at scale. arXiv preprint arXiv:2406.19280.

---

> > ### Comment · Reviewer_GiWs · 2025-11-24
> > **reply to rebuttal**
> >
> > Thanks for the reply and the experiments.  From the table, it is more convincing that larger models do not prefer reasoning, as the performance degrades on many benchmarks.  Also, I am not convinced that demonstrating the reasoning capability of a 3B model is novel enough, as there are many similar papers, some with larger model sizes.
> >
> > Thanks for your honesty on new experiments but I decided not to change my rating.

---

### Official Review · Reviewer_dFCo · 2025-10-30

**Soundness:** 2
**Presentation:** 3
**Contribution:** 2
**Rating:** 4
**Confidence:** 4

**Summary:**

This paper proposes a new training method for medical multimodal LLMs, called InfiMed. The training strategy consists of two steps: (1) supervised fine-tuning with reflective patterns on both general and medical multimodal inputs, resulting in the model InfiMed-SFT; and (2) reinforcement learning with GRPO to further improve model performance. In addition, the authors conduct extensive experiments on various medical benchmarks against several baseline LLMs, demonstrating superior evaluation performance. The ablation studies further validate the effectiveness of each module

**Strengths:**

- The paper is well-structured and clearly presented.
- The proposed pipeline for building a robust multimodal medical LLM is effective.
- The extensive evaluation results convincingly demonstrate the method’s effectiveness

**Weaknesses:**

- The overall contribution is somewhat incremental, building on DeepSeek and prior multimodal LLM work. Nonetheless, the execution is solid.
- The use of general data in the SFT stage is not novel, as this approach has already been widely applied in LLM training (e.g., BianCangLLM [1]).
- The experimental results may not objectively reflect the superiority of this method, since the open-source models and domain-specific LLMs were not fine-tuned on the same datasets as InfiMed.
- Since InfiMed uses Qwen2.5-VL as its backbone, for fairness, baseline medical LLMs should also be built on Qwen2.5-VL to eliminate improvements that stem solely from the backbone. Otherwise, InfiMed should adopt the same backbone as the baselines.
- In the case studies, some examples lack reflective information. Additionally, it is unclear why the reflection ability seems to diminish after reinforcement learning. Clarification on this point would strengthen the paper.

Reference:

1. Wei, Sibo, et al. "Biancang: a traditional chinese medicine large language model." IEEE Journal of Biomedical and Health Informatics (2025).

**Questions:**

Please see my weakness.

---

> ### Author Response · Authors · 2025-11-24
>
> **Q1:** The contribution may appear incremental given prior LLM/MLLM works.
>
> **Ans:** Thanks for the reviewer’s comments. We would like to clarify that our work demonstrates that a strong medical foundation model can be trained with relatively limited data, and we also reveal characteristics that are specific to the medical multimodal domain. For example, we observe that combining general multimodal data with medical textual data improves performance, and that explicit reasoning is not always beneficial. We believe these findings provide useful insights for future research on medical foundation models.
>
> **Q2:** The use of general data in the SFT stage is not novel.
>
> **Ans:** Thanks for the comment. We appreciate the contribution of BianCangLLM to the community. However, BianCangLLM is a text-only model that incorporates general text data mainly to preserve general capabilities, and it does not provide ablation studies to analyze how such data influences model performance. In contrast, our work is situated in the medical multimodal setting, where data scarcity is more severe. We also explicitly show that adding general multimodal data improves performance on medical multimodal benchmarks.
>
> **Q3&Q4:** All comparison MLLMs should use the same backbone or the same dataset.
>
> **Ans:** Thanks for the comment. When training a foundation model, using a specific backbone together with a tailored dataset is a common and widely accepted practice [1-2]. Our goal is to build a strong medical multimodal foundation model, rather than to re-implement each baseline or retrain our own model across multiple different backbones. We choose Qwen2.5-VL because it is one of the strongest openly available MLLMs. Re-training baselines on Qwen2.5-VL or re-training our model on every backbone used by different baselines would not constitute a fair or meaningful comparison, since each baseline is designed and optimized for its own architecture. Such retraining would effectively create new models that no longer represent the originals. Following standard practice, we therefore evaluate baselines in their officially released forms. And our contribution also lies in showing that with limited medical data and appropriate training strategies and reveal domain-specific findings, such as the benefit of mixing general multimodal data with medical text data and the observation that explicit reasoning is not always helpful in medical tasks, which we believe may offer unique insights for the development of future medical foundation models.
>
> **Q5:** It is unclear why the reflection ability seems to diminish after reinforcement learning.
>
> **Ans:** Thank you for the question. The seeming reduction in reflective content after reinforcement learning aligns with the GRPO optimization mechanism. In GRPO, the gradient update objective for each rollout response $o_i$ is normalized by its length $|o_i|$. This means that shorter and correct responses receive a proportionally stronger update signal. As a result, the GRPO process naturally favors concise and accurate answers over longer reflective chains. The reflective data introduced during SFT is primarily intended to provide a broader search space for rollouts during RLVR, rather than to enforce verbose reasoning in the final model. After RL, the model transitions from imitating the format of reflection to producing more efficient outputs, which explains why its generated answers may become more concise compared with the reflection-style responses from SFT.
>
> [1] Xu, W., Chan, H. P., Li, L., Aljunied, M., Yuan, R., Wang, J., et al. (2025). Lingshu: A Generalist Foundation Model for Unified Multimodal Medical Understanding and Reasoning. arXiv preprint arXiv:2506.07044.
>
> [2] Chen, J., Gui, C., Ouyang, R., Gao, A., Chen, S., Chen, G. H., et al. (2024). Huatuogpt-vision, towards injecting medical visual knowledge into multimodal llms at scale. arXiv preprint arXiv:2406.19280.

---

> > ### Comment · Reviewer_dFCo · 2025-11-24
> >
> > Thank you for your reply. However, I find your rebuttal unsubstantiated and perfunctory. In light of this, I have decided to lower my rating.

---

> > > ### Author Response · Authors · 2025-11-24
> > >
> > > Thanks for your follow-up comment. We sincerely apologize if our previous response appeared insufficient or unclear. It was never our intention to provide a perfunctory rebuttal, and we greatly appreciate you taking the time to raise these concerns.
> > >
> > > We would like to declare that the primary objective of our work is to train a competitive medical MLLM using a limited amount of medical data. During this process, we employed techniques such as synthesizing reflective CoT data and uncovered several insightful observations that we believe can provide valuable guidance for the development of future medical MLLMs.
> > >
> > > Moreover, when comparing against existing baseline medical MLLMs, we observed that most of them report only the training backbone without disclosing the full details of their training datasets, such as the specific data sources or the mixing proportions. For example, LINGSHU [1] states that it uses Qwen2.5-VL as its backbone but does not release its detailed training data. Similarly, MedGemma-4B-IT [2] describes the training backbone design but it also includes certain synthetic data that has not been disclosed. Consequently, conducting a strictly controlled comparison is practically infeasible. To further address the reviewer’s concerns, we incorporated a completely new model family, the Gemma3 series, and fine-tuned a Gemma3-4B-IT model using our SFT dataset. The results show substantial improvements over the original Gemma model and achieve SOTA performance within the 3–4B parameter range (excluding comparisons with InfiMed itself). Although this experiment involves only SFT, the outcome clearly demonstrates the effectiveness and efficiency of our SFT data (only 176K samples).
> > > | Model         | MMMU-H&M | VQA-RAD | SLAKE | PathVQA | PMC-VQA | OmniMedVQA | MedXpertQA-MM | Avg.        |
> > > |---------------|----------|---------|-------|---------|---------|------------|---------------|-------------|
> > > |Gemma3 | 34       | 49.89   | 61.17 | 43.21   | 47.94   | 60.85      | 20.85         | 45.41 |
> > > |Gemma3-SFT| 35.33    | 59.86   | 83.33 | 64.74   | 53.27   | 68.69      | 21            | 55.17 |
> > >
> > > Furthermore, during the SFT stage, we not only use general multimodal data and medical text data to improve model performance (as demonstrated in the ablation study section), but we also conducted an additional ablation study on reflective CoT. Specifically, we replaced the reflective CoT data with an equal amount of general CoT data to verify the independent effectiveness of reflective CoT. The corresponding experimental results are presented below.
> > >
> > > | Model                         | MMMU-H&M | VQA-RAD | SLAKE | PathVQA | PMC-VQA | OmniMedVQA | MedXpertQA-MM | Avg.        |
> > > |------------------------------|----------|---------|-------|---------|---------|------------|---------------|-------------|
> > > | InfiMed-SFT-w-generalCoT| 50.67     | 60.53   | 81.94 | 57.67   | 52.78   | 66.36      | 23.4          | 56.19 |
> > > | InfiMed-SFT-3B| 54.7|58.1|82|60.6|53.2|67| 23.5| 57.1 |
> > >
> > >
> > >
> > > [1] Xu, W., Chan, H. P., Li, L., Aljunied, M., Yuan, R., Wang, J., et al. (2025). Lingshu: A Generalist Foundation Model for Unified Multimodal Medical Understanding and Reasoning. arXiv preprint arXiv:2506.07044.
> > > [2] Sellergren A, Kazemzadeh S, Jaroensri T, et al. Medgemma technical report[J]. arXiv preprint arXiv:2507.05201, 2025.

---

### Official Review · Reviewer_Lnrs · 2025-11-01

**Soundness:** 2
**Presentation:** 2
**Contribution:** 2
**Rating:** 4
**Confidence:** 3

**Summary:**

This paper presents the InfiMed-Series models, a family of multimodal large language models designed for medical tasks by augmenting limited medical data with general multimodal and textual reasoning data. It further introduces reflective-pattern-injected chain-of-thought synthesis to enhance the models’ reflective reasoning foundation and enable effective Reinforcement Learning with Verifiable Rewards (RLVR) training.

**Strengths:**

1. This paper is well-written, with a clear and easy-to-follow logical flow.
2. Extensive experiments on multiple benchmarks demonstrate the competitive performance of the proposed InfiMed-Series models.
3. The ablation studies showcase the effectiveness of each component.

**Weaknesses:**

1. Both the SFT and RLVR stages adopt mixed data (such as general multimodal data, textual medical data, and reflective-pattern-injected CoT data) for optimization. What are the proportions of these different data types, and how were they determined? Would varying these ratios lead to different effects? Moreover, why not adopt a progressive approach, feeding different types of data sequentially for optimization?
2. When constructing the reflective-pattern-injected CoT data, how is rejection sampling implemented? Since this process relies on Qwen2.5-VL-32B and Qwen2.5-VL-72B, could it potentially introduce inherent bias?
3. During the RLVR training stage, unlike the SFT stage, the data do not need to contain precise CoTs. However, the RLVR dataset contains only 36K samples, far fewer than the 188K used in the SFT stage. Whether increasing the amount of RLVR training data would further improve model performance remains to be explored.
4. Section 4.4 mentions that step-by-step reasoning introduces redundant steps, increasing the risk of hallucination and degrading performance, whereas Section 4.5 finds that the reasoning process in InfiMed-RL-3B helps identify the correct options. This seems to be contradictory.

**Questions:**

Please refer to the Weaknesses.

---

> ### Author Response · Authors · 2025-11-24
>
> **Q1:** The proportions of different data types and how were they determined?
> **Ans:** We sincerely thank the reviewer for the insightful comments. We illustrated the proportions of the mixed data sources (general multimodal data, textual medical data, and reflective-pattern–injected CoT data) used in both the SFT and RLVR stages in Figure 3. In addition, we conducted four supplementary experiments, where we increased/decreased the amount of general multimodal data and textual medical data by 20k, respectively. The results, summarized in the table, show that adjusting the data ratios leads to different performance variations across benchmarks. Moreover, regarding the reviewer’s suggestion on sequentially feeding different data types, we choose mixed training because it provides a more stable shared representation space and helps avoid potential forgetting when switching data domains.
> Since our focus is on discovering and validating the feasibility and benefits of general multimodal data and medical text data, rather than searching for the optimal ratio. The above results further suggest that there likely exists a more favorable data composition that could enable a model to surpass the current Infimed-SFT-3B under certain tasks. Exploring this optimal ratio, however, is beyond the scope of this work.
> | Model | MMMU-H&M | VQA-RAD | SLAKE | PathVQA | PMC-VQA | OmniMedVQA | MedXpertQA-MM | Avg. |
> |-|-|-|-|-|-|-|-|-|
> | increase 20k general MM data | 54| 60.75| 81.85 | 55.84| 55.49 | 67.52 | 22.3  | 56.82 |
> | decrease 20k general MM data |48| 58.09| 82.13 | 58.37| 51.43 | 66.85| 22.55| 55.34 |
> | increase 20k medical text data |50.67| 63.41| 80.37 | 57.25| 54.42|67.33| 21.4  | 56.41 |
> | decrease 20k medical text data |50.67| 60.31| 82.23 | 58.16| 53.85 | 67.35 | 22.6 | 56.45 |
> | InfiMed-SFT-3B| 54.7|58.1|82|60.6|53.2|67| 23.5| 57.1 |
>
> **Q2:** How is rejection sampling implemented? And could the CoT generation process potentially introduce inherent bias?
> **Ans:** In our pipeline, Qwen2.5-VL-32B-Instruct first generates 10 candidate responses. We then filter out all incorrect responses based on ground-truth answers. For each remaining correct response, Qwen2.5-VL-72B-Instruct is used to score the candidates from multiple dimensions. This multi-stage filtering and scoring procedure helps reduce part of the potential bias introduced by a single model. However, we acknowledge that some degree of inherent bias is unavoidable, as the construction of synthetic medical data inevitably involves large model generation. Therefore, our data processing flow focuses on minimizing the impact of such bias.
>
> **Q3:** Whether increasing the amount of RLVR training data would further improve model performance remains to be explored.
> **Ans:** Thank you for the reviewer’s question. In the medical multimodal domain, the primary bottleneck is the scarcity of high-quality medical multimodal data. Most available data have already been fully utilized in the SFT stage, which makes it difficult to scale the RLVR dataset in the same way as general-domain multimodal settings. Despite this limitation, our model already achieves strong performance with a relatively small amount of RLVR data, suggesting that the current scale is reasonably effective.
> To further examine the reviewer’s concern, we additionally constructed a larger RLVR dataset by adding 15K medical questions (from SLAKE5K, PathVQA9K, and VQA-RAD1.5K). These questions were selected by prioritizing items for which InfiMed-SFT-3B produced the lowest correct counts across ten attempts. The experimental results show that simply expanding the number of RLVR questions does not consistently lead to performance improvements. This suggests that while further increasing RLVR data may still yield some gains, the marginal improvement is likely limited relative to the substantial cost of constructing high-quality medical multimodal RL data.
> | Model| MMMU-H&M | VQA-RAD | SLAKE | PathVQA | PMC-VQA | OmniMedVQA | MedXpertQA-MM | Avg.     |
> |-|-|-|-|-|-|-|-|-|
> | InfiMed-SFT-3B| 54.7|58.1|82|60.6|53.2|67| 23.5| 57.1 |
> | InfiMed-RL-3B | 55.3 |60.5|82.4|62|58.7|71.7|23.6| 59.2|
> | RL+SLAKE5KPATH9KRAD2K| 56| 60.97| 82.23 | 61.49| 58.41| 69.95 | 23.35 | 58.92 |
>
> **Q4:** The conclusions of Section 4.4 and Section 4.5 seem to be contradictory.
> **Ans:** Thanks for pointing this out. The observations in Section 4.4 and Section 4.5 are not contradictory. Section 4.4 shows that in many medical benchmarks, explicit step-by-step reasoning may add noise and sometimes perform worse than answering directly. Section 4.5 highlights a different point: after RLVR, the model can produce higher-quality reasoning when it chooses to reason. This does not imply that reasoning is always helpful, only that RLVR improves the reasoning ability itself.  Therefore, the two sections describe complementary observations rather than a contradiction.

---

### Author Response · Authors · 2025-12-02

Dear Reviewer,

We thank you for your time and efforts in reviewing our work, and we are happy that we were able to address the reviewer’s concerns. In response to your valuable feedback, we have made the necessary revisions and updates to our PDF document. We are grateful for your service to the community.

Regards,
Authors

---

### Note · Authors · 2026-01-06

**Comment:**

Dear Reviewer,

We have carefully considered the reviewers' comments. But we have decided to withdraw the submission. We are grateful for the constructive feedback provided by the reviewers.

Regards,
Authors

**Withdrawal Confirmation:**

I have read and agree with the venue's withdrawal policy on behalf of myself and my co-authors.